# Deep-learning deconvolution and segmentation of fluorescent membranes for high-precision bacterial cell-size profiling

Octavio Reyes-Matte ®[1] ✉, Carsten Fortmann-Grote[1], Beate Gericke[1], Nadja Hüttmann[1], Nikola Ojkic[2] &
Javier López-Garrido ®[1] ✉

Evolutionary studies in bacteria have emphasized genetic and metabolic diversity, while cell-size variation has received less attention. Here we introduce MEDUSSA, a high-throughput method for precise bacterial cell-size profiling based on automatic segmentation of fluorescent membrane images, well suited to studying cell-size diversity. The approach uses deep-learning-based membrane deconvolution, segmentation models fine-tuned for fluorescent-membrane images, and error-corrected cell measurement to extract accurate sizes from individual bacterial cells regardless of shape, size, chaining, or clustering. Our method overcomes limitations of phase-contrast segmentation, yielding reliable single-cell dimensions. We applied MEDUSSA to six strains of *Priestia megaterium* and found over twofold differences in cell volume across strains, largely driven by differences in cell width. We identified a partially-functional PBP1 allele that underlies the reduced width of one strain. Together, these results demonstrate the power of comparative analyses in bacterial cell biology and expand the toolkit to investigate the evolution of bacterial cell size.

Bacterial cells are often perceived as uniformly small, which has led to a relative lack of attention to the evolution of bacterial cell size. Yet, careful microscopic observation has revealed an incredible diversity of bacterial shapes and sizes, with cell volumes spanning multiple orders of magnitude[1,2]. Even among closely related species, cell size can vary significantly. For instance, strains from two *Bacillus* species, *B. thuringiensis* and *B. pumilus*, can differ in volume by nearly an order of magnitude[3], suggesting that cell size is an evolvable trait capable of changing over relatively short evolutionary timescales.

Despite this diversity, our appreciation of bacterial cell size variation at finer evolutionary scales remains limited. Systematic studies have focused on mutant libraries in a few model strains[4,5]. However, comparative studies analyzing the size of different bacterial strains or species are rare[6,7], and most available data come from independent studies, often using different imaging methods and manual measurements of a few representative cells cultured in different conditions. While this highlights bacterial size diversity, methodological inconsistencies make cross-study comparisons difficult. As a result, we lack a clear picture of how bacterial cell size varies among closely related species and strains.

Recent advances in image analysis pose a unique opportunity for the standardization of methods to measure bacterial cell size and may help to address these challenges. In particular, the incorporation of machine learning into image analysis pipelines has enabled the automatic identification and precise delineation of boundaries of individual bacterial cells in microscopy images. This process, known as image segmentation, generates masks of individual cells that can be used for accurate cell size measurements. The U-Net, a convolutional neural network architecture originally designed for the analysis of biomedical images[8,9], is behind some of the most powerful image segmentation algorithms currently available, such as StarDist[10], DeLTA[11], Cellpose[12], MiSiC[13], Omnipose[14], DeepBacs[15] and DistNet[16], to name a few. More recently, vision transformers[17] have enabled the development of foundation models[18,19], which support strong off-the-shelf performance and facilitate fine-tuning on custom data. Several models have leveraged large public microscopy datasets to achieve high flexibility, including microSAM[20], Cellpose-SAM[21] and CellSAM[22]. These tools are significantly improving the precision and automation of bacterial cell segmentation, paving the way for more consistent and reproducible cell size measurements.

[1]Max Planck Institute for Evolutionary Biology, Plön, Germany. [2]School of Biological and Behavioral Sciences, Queen Mary University of London, London, UK.
✉e-mail: reyesmatte@evolbio.mpg.de; lopezgarrido@evolbio.mpg.de

Here, we have fine-tuned and benchmarked some of these models for segmenting fluorescent membranes. Building on this, we developed a pipeline for extracting precise single-cell size information from fluorescent membrane images, which allows the characterization of individual cell sizes regardless of clustering or chaining patterns. We applied this pipeline, which we call MEDUSSA (MEmbrane DeconvolUtion and Segmentation for Size Analyses), to investigate cell-size diversity across six strains of *Priestia megaterium*, a Gram-positive bacterium commonly used for physiological studies and biotechnological applications[23]. We found significant differences in volume between the strains—spanning more than twofold—underpinned by consistent differences in cell length and width. We further link width differences between two of the strains to a specific mutation in *ponA*, which encodes penicillin-binding protein 1 (PBP1), resulting in a hypomorphic PBP1 allele.

## Results

### Limitations of phase contrast segmentation for single cell studies

Deep-learning-based segmentation algorithms are versatile and can segment different types of images, as long as they are trained with appropriately labeled datasets for the specific task. However, phase-contrast has become the default imaging modality for bacterial segmentation, as it provides excellent contrast between cells and background without the need for staining or genetic manipulation[4,14,16,24–28]. Despite its advantages, phase contrast images have limitations for cell size estimation. Firstly, they lack clear visual references to precisely define cell boundaries (Fig. 1e). Secondly, phase contrast does not allow the distinction of cells that remain attached after division, complicating the identification of individual cells in bacterial species that form cellular chains[29,30]. For instance, *Bacillus subtilis* and related species form heterogeneous populations in which single cells coexist with long chains consisting of dozens of cells[31,32]. Because the separation of individual cells within a chain is not visible in phase-contrast images, state-of-the-art segmentation models optimized for phase-contrast segmentation, such as the Omnipose "bact_phase_omni" model[14], segment chains as very long individual cells, making it difficult to extract information at the single cell level (Fig. 1a,c).

This issue can be circumvented by staining the cell membranes with lipophilic fluorescent dyes, such as FM 4-64[33] or Nile Red[34], which allows the visualization of the septal membranes that separate individual cells within chains (Fig. 1b, d). The fluorescence intensity profile of FM 4-64-stained membranes shows a sharp peak that can be traced around the cell (Fig. 1e, f), providing an unambiguous guide to delimit the cell boundaries. We therefore explored the performance of different segmentation models on fluorescent membrane images.

### High-throughput segmentation of fluorescent membranes

We fine-tuned existing deep-learning segmentation models for segmenting fluorescent membrane images of bacterial cells, and benchmarked their performance. We focused on Cellpose3[35], Omnipose[14], and the SAM-based foundation models microSAM[20] and Cellpose-SAM[21]. Omnipose has been reported to perform better on elongated cells than other segmentation algorithms[14]. The models were fine-tuned using over seven thousand FM 4-64-stained *B. subtilis* cells annotated with JFilament active-contour software (Fig. 1g)[36], which allows the semiautomated generation of snakes that track the fluorescent membrane peak around the cell, thereby diminishing user annotation bias. JFilament snakes precisely delimit cell contours and provide accurate cell dimensions (Supplementary Fig. 1). The training dataset included single cells, cells within chains and cells within clusters (see Methods for details). We tested the fine-tuned models with images of three bacilli of different sizes: *B. subtilis*, *B. thuringiensis*, and *Priestia megaterium* (Fig. 2a; Supplementary Fig. 2a). We also tested Morphometrics, a non-deep-learning segmentation method that has shown good performance on fluorescent membrane images[4]. Cellpose-SAM showed the best performance in benchmarking tests (mean F1 at IoU 0.8, 0.86 ± 0.08), followed by Omnipose (0.76 ± 0.16), Morphometrics (0.64 ± 0.22), Cellpose3 (0.59 ± 0.17) and microSAM (0.30 ± 0.16) (Fig. 2a). Because segmentation

algorithms are known to often fail on elongated cells, we further tested the performance of the fine-tuned models on a *Lysinibacillus* strain that naturally produces long cells under laboratory conditions (Supplementary Fig. 2b). Only Omnipose segmented long cells appropriately. The other models oversegmented them. Since our goal was to develop a pipeline for cell size analysis across different strains and species, many of which may naturally form long cells, we further analyzed Omnipose segmentation to improve its performance.

The fine-tuned Omnipose model produced segmentation masks that lacked aberrations, truncations, and protrusions for isolated cells and individual cells within a chain, maintaining proper rod-shaped morphology. However, it struggled with clustered cells, often producing indented or truncated masks (Supplementary Fig. 2a). We reasoned that this limitation might reflect the broader and less well-defined membrane signal in raw fluorescent images, particularly in densely packed regions[37]. To test whether improving membrane definition could enhance segmentation accuracy, we retrained Omnipose using deconvolved images. Deconvolution reduces out-of-focus blur and produces sharper membranes[38,39]. When evaluated on deconvolved membrane images, the retrained model, FMSeg_Omni (hereafter FMSeg), performed better than Omnipose retrained and evaluated on the corresponding raw membrane images, and reached a performance better than that of Cellpose-SAM retrained and evaluated on raw membrane images (mean F1 score at IoU 0.8, 0.88 ± 0.09) (Fig. 2a). More broadly, deconvolution affected model performance in a model-dependent manner (Fig. 2a; Supplementary Fig. 2; Supplementary Fig. 3): Omnipose, microSAM and Morphometrics performed better on deconvolved membrane images than on raw membrane images, whereas Cellpose-SAM and Cellpose3 showed the opposite trend, despite being fine-tuned separately for each image type.

FMSeg successfully identified and segmented single cells within groups, isolated cells, chains, and even extremely elongated cells, regardless of the species or size (Fig. 2b, c; Supplementary Fig. 3). Additionally, FMSeg successfully segmented yeast and cocci of various sizes, both isolated and clustered, and worked with alternative membrane labels, including Mito-tracker Green and GFP fused to a homogeneously distributed membrane protein (QoxB) (Fig. 2d; Supplementary Fig. 4).

To facilitate the segmentation of fluorescent membrane images acquired with different microscopes, including systems without deconvolution capabilities, we trained an image restoration model to predict deconvolution from non-deconvolved images (FM2FM model; Fig. 3a). When direct deconvolution of membrane images is possible, it can be used directly. FM2FM provides an alternative for datasets in which this is not feasible. FM2FM was trained using the CARE framework[40], in which a supervised U-Net predicts pixelwise fluorescence values while accounting for spatial context (i.e., neighboring pixel intensities). The fluorescence profiles of the predicted images closely matched those of deconvolved images (Fig. 3a–c), and the model could generate images amenable to segmentation from different microscopes (Supplementary Fig. 5).

To assess whether deconvolution prediction affects segmentation, we compared cell size parameters—width, length, cross-sectional area, surface area and volume (see next section and methods for details)—extracted from masks generated using deconvolved and FM2FM-predicted images. We used multiple size metrics to verify that information relevant to downstream analyses is preserved across processing scenarios, as single metrics often fail to predict how image-derived measurements propagate into downstream results[41]. We also compared shape descriptors unrelated to cell size, such as convex hull area, eccentricity and solidity. The distributions were very similar, with no statistically significant differences (Fig. 3d), indicating that membrane deconvolution predictions from raw membrane images has a minimal effect on segmentation and does not impact cell size calculations.

Additionally, we trained a fluorescence translation model to infer deconvolved membranes from cytoplasmic fluorescence (FP2FM model; Supplementary Fig. 6), which might be useful when membrane staining is not feasible. Previously developed CARE restoration models for predicting superresolution membranes can also be used[15]. Although these approaches

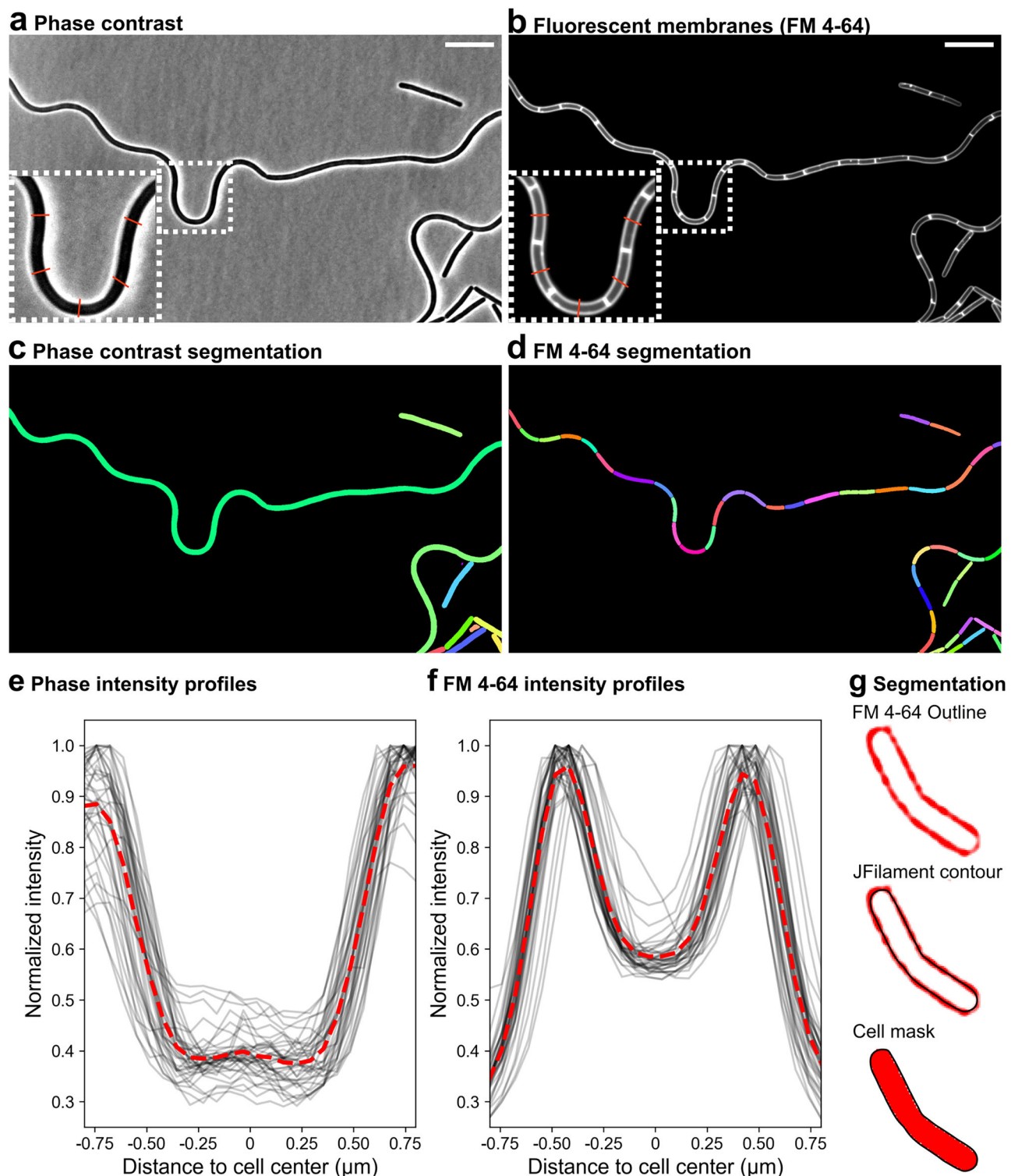

**Fig. 1 | Phase-contrast versus fluorescent-membrane segmentation. a,b** Phase-contrast (**a**) and fluorescent membrane images (**b**) of *B. subtilis* growing vegetatively. Membranes were stained with FM 4-64. The insets show zoomed-in views of the boxed regions; the red lines illustrate the paths along which intensity profiles are calculated in (**e** and **f**). **c,d** Segmentation masks for the phase-contrast (**c**) and fluorescent membrane (**d**) images shown in (**a** and **b**). The phase-contrast image was automatically segmented with Omnipose[14] and the fluorescent-membrane image was manually segmented with JFilament[36] (see also **g**). Every mask has a different color. **e,f** Phase-contrast (**e**) and fluorescent membrane (**f**) intensity profiles of 39 cells, measured along a 24-pixel line perpendicular to their long axis, centered at midcell (illustrated in insets in (**a** and **b**). Pixel intensities were normalized to the maximum intensity observed across the lines for each imaging modality. **g** Example of manual fluorescent-membrane segmentation using the active contour software JFilament[36].

## a Performance quantification of segmentation models

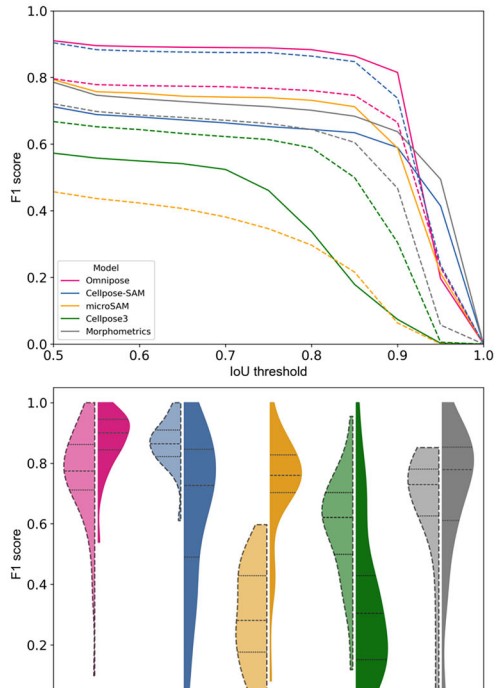

## b Test data example

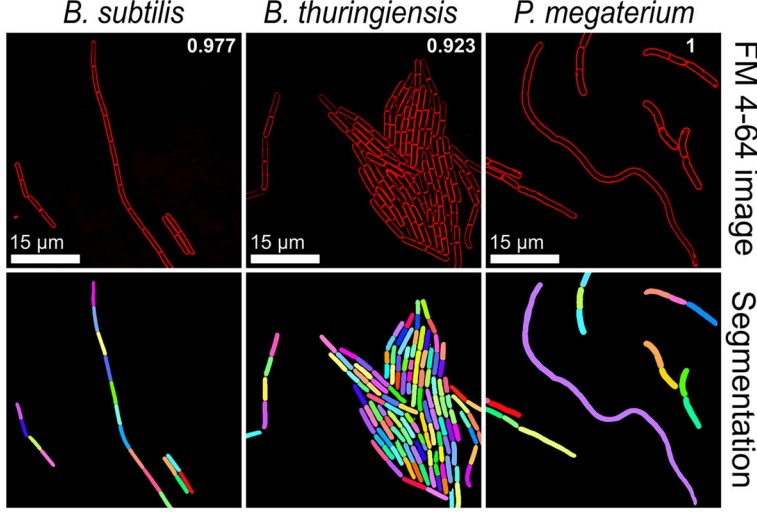

## c Segmentation in a complex image (*B. subtilis*)

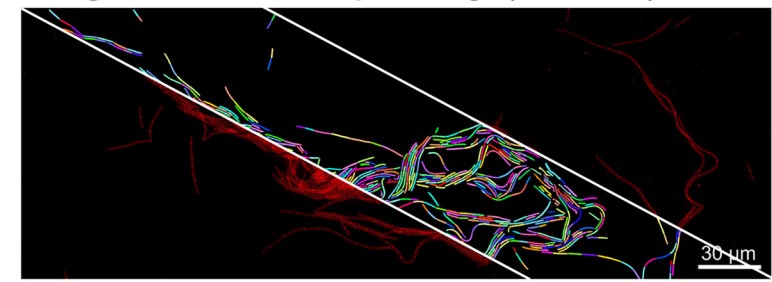

## d Segmentation of different cell types and membrane dyes

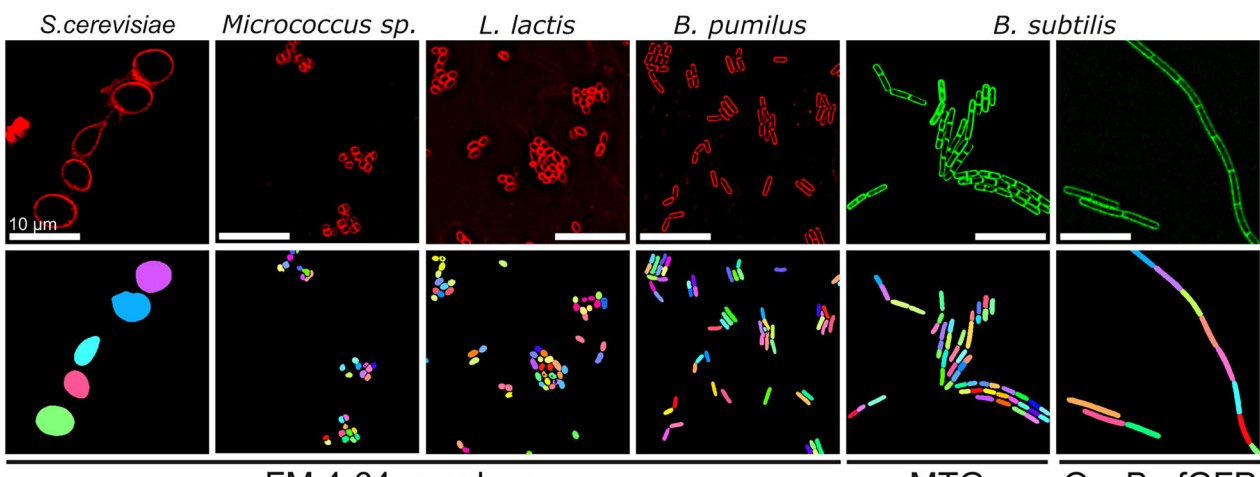

expand the applicability of the method, they may introduce biases in cell size measurements (Supplementary Fig. 6).

Altogether, our fine-tuned segmentation and deconvolution prediction models offer a robust approach for the high-throughput segmentation of individual bacterial cells with fluorescent membranes, regardless of their size, shape and chaining pattern.

### MEDUSSA, an analytical pipeline for extracting cell size from fluorescent membrane images

We designed an algorithm to extract precise 2D cell dimensions—width and length—and extrapolate 3D measurements, such as membrane surface area and volume (Fig. 4a). Starting from FMSeg-generated masks, we applied a distance transform to compute, for each pixel, the shortest distance to the mask periphery. We then overlaid a medial skeleton on this transform and sampled the radius along the skeleton to obtain cell width across the long axis. Radii at the skeleton endpoints were used to estimate the size of the hemispherical caps, yielding length and width profiles for each cell. Assuming rotational symmetry and imaging in the medial focal plane, we estimated volume and surface area by summing cylindrical segments along the skeleton and adding hemispherical ends. Additional details are provided in the Methods section.

**Fig. 2 | Evaluation of models for fluorescent-membrane segmentation.**
**a** Quantitative benchmarking of Omnipose (pink), CellposeSAM (blue), microSAM (yellow), Cellpose3 (green) and Morphometrics (gray) on fifty-six images of *B. subtilis*, *B. thuringiensis* and *P. megaterium*. Omnipose, CellposeSAM, microSAM, and Cellpose3 were fine-tuned separately on raw and deconvolved fluorescent membrane images; models fine-tuned on raw images were evaluated on raw images, whereas models fine-tuned on deconvolved images were evaluated on deconvolved images. Top, F1 score across Intersection over Union (IoU) thresholds. Lines are the mean F1 scores of 56 images. Dashed lines correspond to model performance on raw fluorescent membrane images, and solid lines on deconvolved images. Bottom, F1 at IoU of 0.8. The first violin plot of each pair (lighter color and delineated by a dashed line) corresponds to model performance on raw images, and the second violin plot (darker color) to model performance on deconvolved images. The horizontal dashed line in the violin plots indicate the median, and the dotted lines the first and third quartiles. We called Omnipose model fine-tuned on deconvolved membranes

FMSeg_omni (FMSeg) **b** Examples of images showing chains, clusters, and heavily elongated cells. Top, deconvolved micrographs of *B. subtilis*, *B. thuringiensis* and *P. megaterium* cells stained with FM 4-64; bottom, FMSeg masks. F1 scores at IoU 0.8 are indicated at the top-right corner of each image. Scale bar, 15 μm. **c** FMSeg segmentation of a *B. subtilis* microcolony. The image is split into three sectors. Segmentation results are shown in the middle sector, and FM 4-64 stained membranes in the flanking sectors. Scale bar, 30 μm. **d** Segmentation of cells of different shapes and sizes, with membranes labeled with FM 4-64 or other fluorophores. Top, deconvolved fluorescent images; bottom, FMSeg segmentation. *Saccharomyces cerevisiae*, *Micrococcus sp.*, *Lactococcus lactis* and *B. pumilus* cells were stained with FM 4-64 (red). *B. subtilis* was stained with MitoTracker Green (MTG). We also imaged a *B. subtilis* strain producing a GFP fused to the membrane protein QoxB (QoxB-sfGFP, green), which is homogenously distributed in the membrane. See Supplementary Fig. 4 for benchmarking of MTG and QoxB-GFP segmentation. Scale bars, 10 μm.

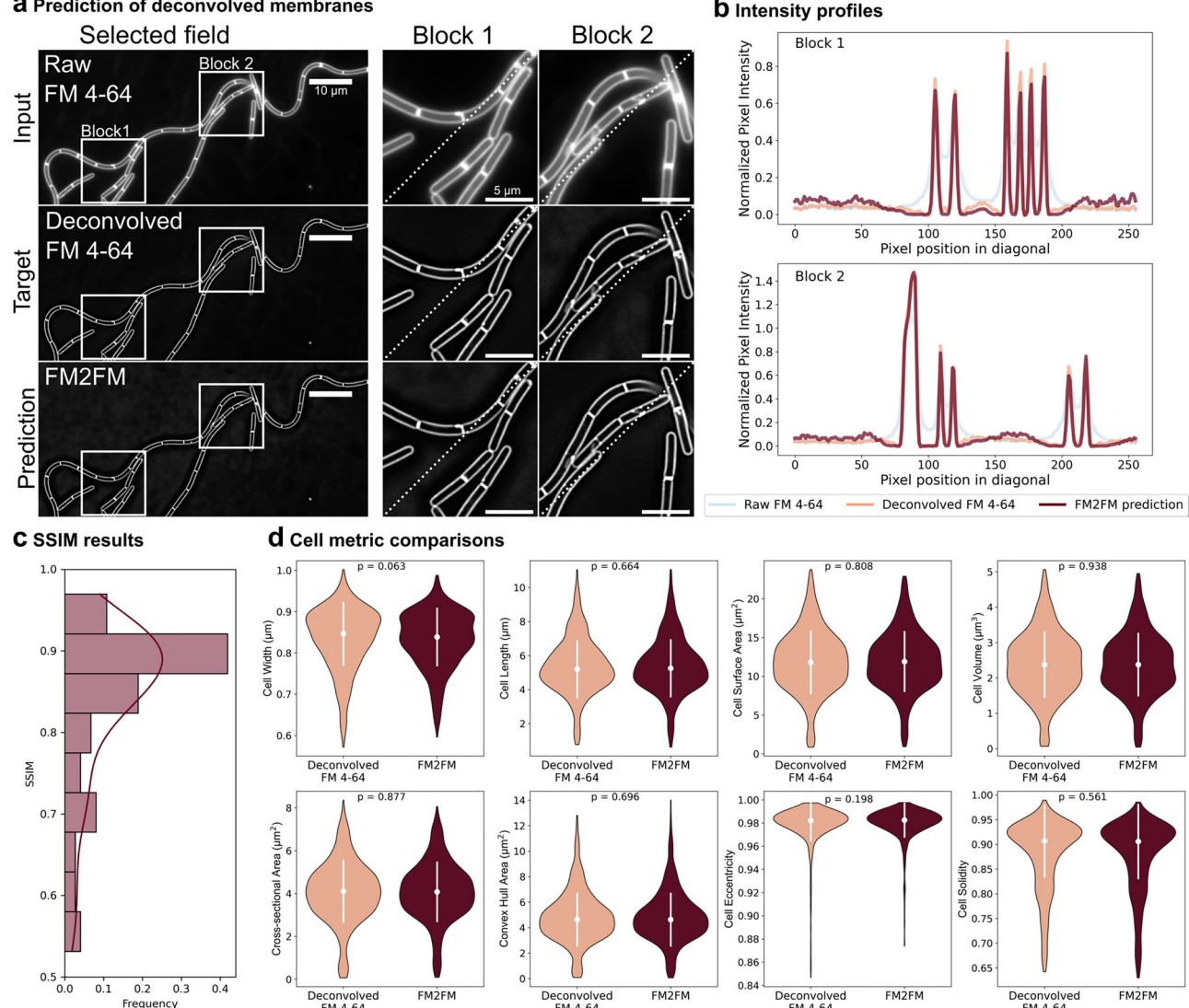

**Fig. 3 | Deconvolution prediction. a** Example of deconvolution prediction with FM2FM. Top, input raw fluorescent membrane images (Raw FM 4-64); middle row, true deconvolved fluorescent membrane images (Deconvolved FM 4-64); bottom, FM2FM-predicted image. White squares mark regions zoomed in at right (Block 1 and Block 2). Diagonal dotted lines indicate profile traces in (**b**). Scale bars: full images, 10 μm; zoomed-in blocks, 5 μm. **b** Fluorescence profiles across the dotted lines in Block 1 (top) and Block 2 (bottom). Light blue, raw FM 4-64; orange, deconvolved FM 4-64; burgundy, FM2FM prediction. **c**, Structural similarity index measure (SSIM) between FM2FM predicted images and deconvolved images. The

distribution of SSIM values across 74 image crops is shown. **d** Violin plots of cell width, length, surface area, volume, cross-sectional area, convex hull area, eccentricity and solidity calculated from deconvolved FM 4-64 images (orange) and FM2FM-predicted images (burgundy). The white dots and vertical lines within the violin plots represent the medians and standard deviations. P values from statistical comparisons between distributions (ANOVA or Kruskal test; see methods for details) are indicated in each panel. Twenty images (over 1000 cells) were processed and segmented with FMSeg. See the Methods for size calculation details.

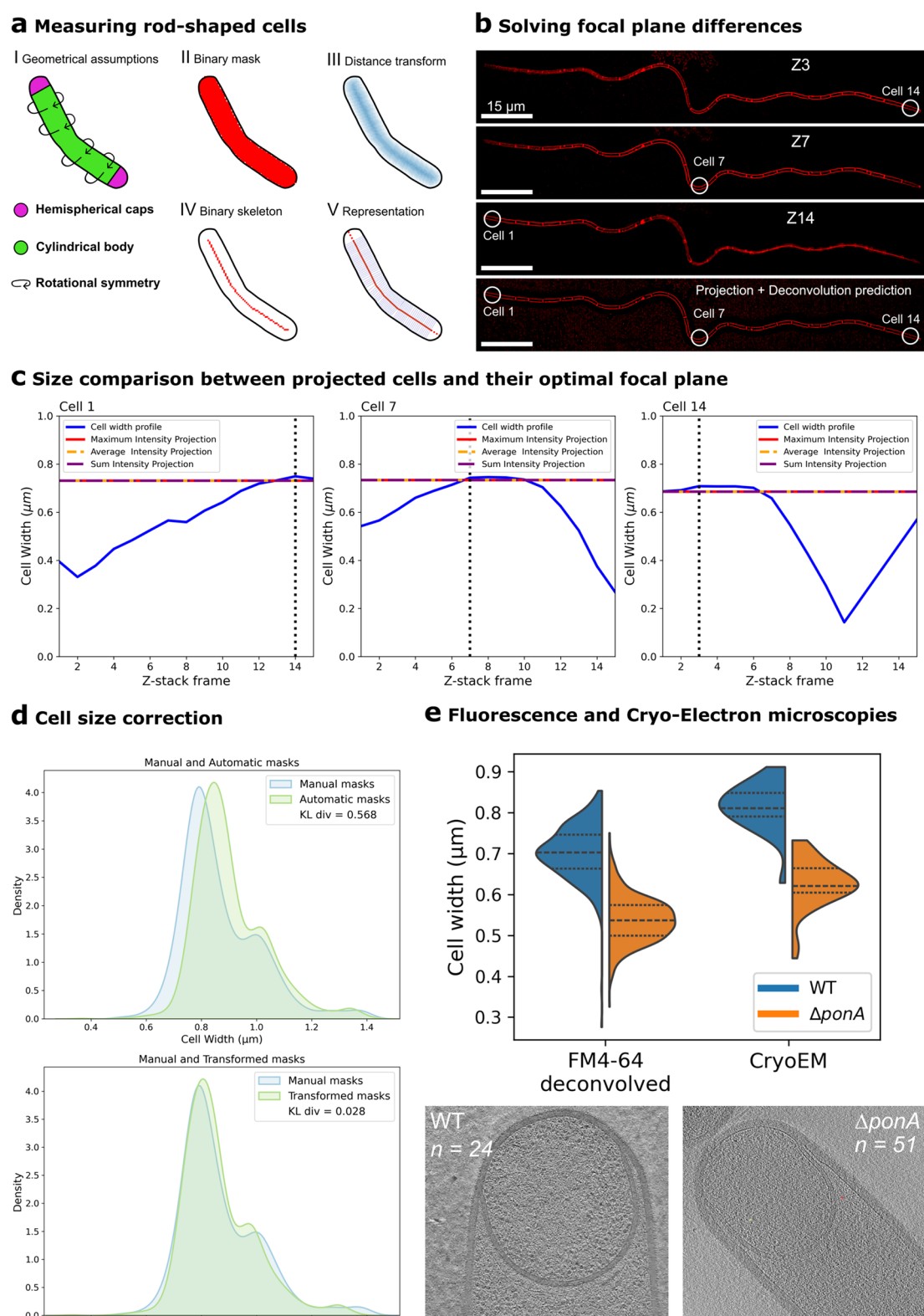

**a** Measuring rod-shaped cells

I Geometrical assumptions  II Binary mask  III Distance transform

- Hemispherical caps
- Cylindrical body
- Rotational symmetry

IV Binary skeleton  V Representation

**b** Solving focal plane differences

15 μm  Z3  Cell 14

Z7  Cell 7

Cell 1  Z14

Cell 1  Projection + Deconvolution prediction  Cell 7  Cell 14

**c** Size comparison between projected cells and their optimal focal plane

Cell 1 / Cell 7 / Cell 14

— Cell width profile
— Maximum Intensity Projection
— Average Intensity Projection
— Sum Intensity Projection

**d** Cell size correction

Manual and Automatic masks — Manual masks, Automatic masks, KL div = 0.568

Manual and Transformed masks — Manual masks, Transformed masks, KL div = 0.028

**e** Fluorescence and Cryo-Electron microscopies

WT / ΔponA

FM4-64 deconvolved / CryoEM

WT n = 24 / ΔponA n = 51

Integrating segmentation with size calculations revealed two key factors critical for reliable measurements (Fig. 4b–d; Supplementary Fig. 7; Supplementary Fig. 8). The first is that the size of the segmentation masks is sensitive to the focal plane (Fig. 4b, c; Supplementary Fig. 7). For rod-shaped cells, width is maximal at the mid-cell focal plane but decreases above and below it (Fig. 4c)[4]. It is therefore critical to ensure that size measurements are

taken from mid-cell focal planes. However, due to minor aberrations in microscopy slides, cells within a field of view often reside in slightly different focal planes (Fig. 4b; Supplementary Fig. 7). To address this, we acquired z-stacks and projected them onto a single plane before predicting deconvolution with FM2FM and segmentation (Fig. 4b). This approach yielded size estimations equivalent to those obtained from mid-cell focal planes

**Fig. 4 | Cell size estimation from fluorescent-membrane segmentation masks.**
**a** Cell size calculation workflow. **b** Representative z-slices from an optical-sectioning stack of a 14-cell *B. subtilis* chain stained with FM 4-64. Different cells are in focus in different slices. Circled cells are analyzed in (**c**). The last image is a maximum-intensity projection of whole stack, processed FM2FM, where all cells appear in focus. Scale bar, 15 μm. Full stack in Supplementary Fig. 3. **c** Width profiles (solid blue lines) of the cells circled in (**b**) on different z-slices across the optical sectioning stack: Cell 1 (right graph), Cell 7 (middle graph) and Cell 14 (right graph). Width is maximized on the slice in which the cell is in focus (vertical dotted lines). Cell width calculated from projected images is indicated by horizontal dashed lines. Results were equivalent regardless of the projection type (maximum intensity, red dashed line; average intensity, orange dashed line; sum, purple dashed line). Cells were segmented using JFilament in every z-slice and projections with FMSeg. **d** FMSeg masks are slightly wider than manual JFilament masks. Density plots show the width distribution of 1100 bacilli cells. The top plot compares JFilament masks (blue) to FMSeg masks (green). After transforming the data with our sampling strategy (transformed masks, bottom plot), the distributions align better. The similarity of the distributions was quantitatively compared using Kullback–Leibler Divergence (KL div, values in insets within the graphs). See Supplementary Fig. 8 and the Methods section for details. **e** Widths of wild-type (WT, blue) and Δ*ponA* (orange) cells from sporulating *B. subtilis* cultures, measured from deconvolved fluorescent-membrane images segmented with FMSeg (FM 4-64 deconvolved; 326 and 617 cells for WT and Δ*ponA*, respectively), or from CryoEM images (24 and 51 cells for WT and Δ*ponA*, respectively). Dashed lines mark the median: WT 0.70 μm (FM) vs 0.81 μm (cryo-EM); Δ*ponA* 0.54 μm (FM) vs 0.63 μm (cryo-EM). FM widths were ~14% lower than cryo-EM, but the WT/Δ*ponA* ratio was nearly identical (1.300 vs 1.304; <1% difference). Cryo-EM images courtesy of E. Tocheva (Δ*ponA*) and E. Villa & K. Khanna (WT).

(Fig. 4c), enabling accurate measurements across entire microscopy field, regardless of focal plane differences.

Second, we observed that the FMSeg model consistently generated masks slightly wider than ground truth masks (Fig. 4d; Supplementary Fig. 8a). This bias propagated into surface area and volume calculations (Supplementary Fig. 8b). Fortunately, the overestimation was independent of cell size, allowing us to apply a systematic correction (Fig. 4d; Supplementary Fig. 8b). We developed a size correction pipeline that allowed us to predict the "true" cell size based on the dimensions of FMSeg-generated masks (Supplementary Fig. 8; see methods for details). We validated this approach by correcting the segmentation outputs for *B. subtilis*, *B. thuringiensis* and *P. megaterium* cells used in our benchmarking tests (Fig. 2a). As shown in Fig. 4d, FMSeg-mask width distribution was slightly shifted compared to that obtained from ground-truth masks generated using JFilament. After applying the correction, both distributions became nearly identical (Fig. 4d).

To assess the accuracy of our cell size measurements, we compared estimates from our pipeline with those obtained from plunged-frozen *B. subtilis* imaged by cryo-electron microscopy (cryoEM), a technique that provides high-resolution images of cells preserved in their native physiological state. We focused on sporulating *B. subtilis* cells, as we could access sets of cryoEM images of these cells, kindly provided by Elitza Tocheva[42], Elizabeth Villa and Kanika Khanna[3,43,44]. To test robustness across different cell sizes, we measured both wild-type cells and cells lacking PBP1 (Δ*ponA*), which are ~25% thinner than wild type cells[45]. Our pipeline reproduced the difference in width between wild-type and Δ*ponA* cells observed by cryo-EM, although the absolute width estimates were on average ~14% lower than those obtained by cryo-EM (Fig. 4e). This discrepancy may reflect a systematic bias in our fluorescence-based measurements, but it could also arise from differences in sample preparation, imaging modality, or growth conditions between the two datasets. These results indicate that our pipeline captures biologically relevant differences in cell width and yields absolute size estimates that are broadly consistent with cryo-EM.

Overall, our fine-tuned segmentation model, coupled with deconvolution prediction and size correction, enables high-throughput, reliable cell size measurements from fluorescent-membrane microscopy images. We refer to the full pipeline of deconvolution prediction, segmentation and cell size measurement as MEDUSSA (MEmbrane DeconvolUtion and Segmentation for Size Analyses) (Supplementary Fig. 9).

### Insight into intraspecific cell-size diversity in *P. megaterium*

We applied MEDUSSA to study variation in cell size across strains of *P. megaterium*[46], a member of the *Bacillaceae* generally considered to produce large cells, but for which intraspecific morphological diversity has been reported[23,47,48]. We focused on six strains widely used in physiology and biotechnology: the type strain DSM 32 (ATCC 14581)[49] from the German Collection of Microorganisms and Cell Cultures (DSMZ); the classic multiplasmid strain QM B1551[50]; the naturally plasmidless strain DSM 319[50,51]; KM, used in early bacteriophage studies; 899, also used in early

bacteriophage and bacteriocin studies; and WH320[52], a Lac⁻ derivative of DSM 319 commercially available for protein-expression. We sampled cultures of each strain during exponential and stationary phases in LB medium (Supplementary Fig. 10), imaged them in agarose pads supplemented with FM 4-64 for membrane staining, and processed the images through our complete pipeline to obtain estimations of the sizes of individual cells (Fig. 5; Supplementary Fig. 11a). Although all strains tended to form chains (Fig. 5a), MEDUSSA could resolve single cells with high precision. Single-cell dimensions were generally consistent across experimental runs, although we observed variation in population averages of some strains (up to 27% for cell length and 12% for cell width; Supplementary Fig. 11a), likely due to day-to-day differences in conditions outside experimental control. Technical variation among samples from the same culture was low (2-3% at most for cell width and length) (Supplementary Fig. 11b).

We observed striking between-strain diversity in cell width: during exponential growth, median width ranged from 0.92 μm in WH320 to 1.39 μm in 899 (Fig. 5b; Supplementary Fig 11a). Most strains were slightly thinner in stationary phase, except KM, whose width distribution broadened without a clear shift in the median (Fig. 5b). Although width showed a smaller coefficient of variation (CV) than length—consistent with tighter control—the width CVs of *P. megaterium* strains were generally higher than those reported for *B. subtilis*[5,53–55]; ranging from 0.05 to 0.081, also reproduced by us in Supplementary Fig. 11a), with some strains reaching ~0.15 in exponential or stationary phase (Supplementary Fig. 11a). Notably, cells of different widths were sometimes present within a single chain (arrowheads in Fig. 5a).

We also observed significant differences in cell length across strains: in the exponential phase, the median ranged from 4.06 μm in DSM 32 to 5.76 μm in strain 899, and cells shortened in the stationary phase, as expected (Fig. 5a,c). Within-strain variation in cell length was high in some cases (Fig. 5c; Supplementary Fig. 11a), with rare, extremely elongated cells occurring in several strains; the more dramatic cases were in DSM 32, where approximately 10% of the cells were over 10 μm long (Fig. 5c, inset). These extreme elongation events were not observed in the stationary phase (Fig. 5c), suggesting that it is a phenomenon associated with active growth and that these filaments septate into multiple cells or lyse upon entry into the stationary phase.

These width and length differences underpinned differences in surface-to-volume ratios—thinner cells exhibited higher surface-to-volume ratios than thicker cells[56,57]—and contributed to the diversity of cell volumes observed between strains, spanning nearly threefold (Fig. 5d, e).

### *P. megaterium* WH320 carries a partially functional version of PBP1

Most accounts of *P. megaterium* cell width report values of ~1.2-1.5 μm[23]. However, two of our strains, DSM 32 and WH320, had widths below these values. We were particularly surprised by WH320, whose median width during exponential growth was ~0.9 μm, as this strain is a derivative of DSM 319, one of the thickest strains in our panel (median width in exponential phase ~1.4 μm).

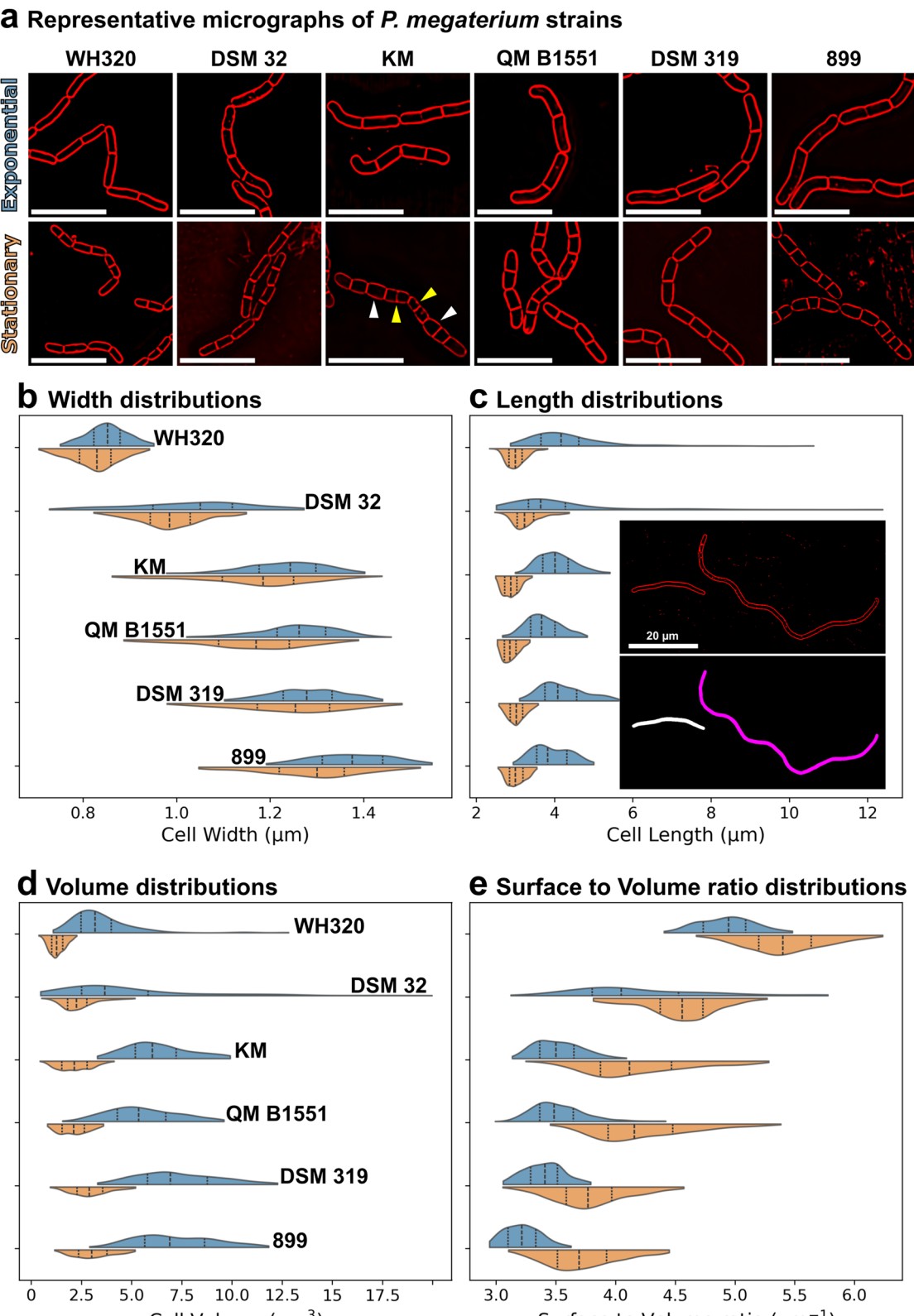

**a** Representative micrographs of *P. megaterium* strains

**b** Width distributions

**c** Length distributions

**d** Volume distributions

**e** Surface to Volume ratio distributions

WH320 was generated by ethyl-metanosulfonate (EMS) mutagenesis of DSM 319, selecting for Lac⁻ clones[52]. One potential explanation for the reduced cell width of WH320 is that an additional mutation introduced during EMS mutagenesis or at some point after the two strains diverged affected cell width. To explore this possibility, we sequenced the genome of WH320 and compared it to the parental DSM 319 reference genome[50]. The two strains differed by 37 single-nucleotide polymorphisms, with the majority of them being G·C→A·T transitions, consistent with EMS mutagenesis[58] (Supplementary Data in Zenodo repository). We also observed other mutations, including transversions and indels, which might have occurred during transfer of the strains in the lab before we acquired them. One of the transitions caused a glycine-to-arginine amino acid

**Fig. 5 | Morphological characterization of six *P. megaterium* strains.**
**a** Fluorescence micrographs of FM 4-64-stained cells of different *P. megaterium* strains sampled form exponential and stationary phase LB cultures: WH320, DSM 32, KM, QM B1551, DSM 319 and 899. Arrowheads in the KM stationary image point to thicker (white) and thinner (yellow) cells within a chain. Scale bars, 10 μm. **b**–**e** Distributions of cell with (**b**), length (**c**), volume (**d**) and surface to volume (**e**) for the indicated strains (strain names in (**b** and **d**); same order maintained in the other panels) during exponential (blue) and stationary (orange) phase. The dashed line in the violin plots indicate the median, and the dotted lines the first and third quartiles.

The inset in panel **c** shows a micrograph of a long cell produced by DSM 32 during exponential growth (membranes, red; scale bar, 20 μm). The following number of cells were analyzed for each strain in exponential and stationary phase: WH320, 153 and 961 cells; DSM 32, 192 and 904 cells; KM, 149 and 856 cells; QM B1551, 292 and 447 cells; DSM 319, 195 and 471 cells; 899, 758 and 548 cells. The central 95% of observed values are represented. See Supplementary Fig. 11 for comparison between experimental runs. Prior to measurement, segmentation output was manually curated to remove aberrant masks and truncated masks at image edges.

substitution in the catalytic site of *P. megaterium* β-galactosidase, likely responsible for the Lac⁻ phenotype. We also found a C to A transversion in *ponA* that produced an alanine-to-aspartic acid change at residue 214 in the transglycosylase domain of the class A penicillin binding protein PBP1 (Supplementary Data in Zenodo repository).

As noted above, PBP1 modulates cell width in *B. subtilis*, with *ponA* mutants being ~25% thinner than the wild type and *ponA* overexpression producing cells thicker than the wild type (Fig. 4e)[45,59]. We therefore hypothesized that the *ponA* mutation reduces PBP1 activity and underlies the thin phenotype of WH320. To get insight into this possibility, we asked whether the *ponA* alleles from DSM 319 and WH320 could complement a *B. subtilis ponA* mutant when produced ectopically. We placed both alleles under the IPTG-inducible promoter $P_{spank}$ at the ectopic *amyE* locus, and measured cell width after incubation with varying IPTG concentrations. As shown in Fig. 6, expression of DSM319 *ponA* produced a marked increase in cell width, even at low IPTG concentrations. In contrast, expression of WH320 *ponA* caused a milder increase in cell width and never reached the values observed with $ponA_{DSM319}$, even at high induction levels. These results indicate that the alanine-to-aspartate substitution in *ponA* reduces PBP1 function.

## Discussion

Here, we introduce MEDUSSA, an image-analysis pipeline that extracts accurate single-cell size measurements from fluorescent-membrane microscopy images, and demonstrate its utility for dissecting mechanisms of cell-size diversity among bacterial strains. The workflow combines deep-learning–based image restoration for deconvolution prediction with automated segmentation to delineate cell boundaries from membrane signal, followed by mask analysis to estimate cell size. Because each step is modular, the restoration (FM2FM), segmentation (FMseg), and measurement components can be used independently to suit specific study needs (Supplementary Fig. 9). Applied across strains and species, MEDUSSA yields cell-size measurements that can be further interrogated to identify underlying mechanisms.

MEDUSSA is well-suited for comparative studies because it addresses key shortcomings of previous methods. First, fluorescent membrane imaging enables precise delineation of cell boundaries by tracking the peak membrane signal around each cell (Fig. 1). This supports consistent training-set annotation and yields more reliable cell-size estimates than phase-contrast-based approaches[37] (Fig. 1). Second, membrane dyes reveal septal membranes, allowing clear separation of daughter cells that remain attached after division and thus enabling reliable length measurements for both isolated cells and chains—something that is difficult with phase contrast (Fig. 1)[30]. Third, the pipeline corrects focal-plane differences among cells within the same image, which would otherwise bias measurements because focal-plane shifts alter mask size (Fig. 4b-c)[4]. Specifically, projecting fluorescence images from a z-stack and applying a deconvolution-prediction restoration algorithm (FM2FM) produces segmentation-ready images that yield mid-focal-plane masks. Thus, this standardized protocol of restoration, segmentation, and measurement converts membrane-fluorescence images into population-level size distributions.

A trade-off is the need for fluorescent labeling. Several membrane and cell-wall dyes are routinely used in bacterial cell biology[33,60–65]. Most experiments presented here use FM 4-64, a lipophilic styryl fluorophore that associates with the outer leaflet of the cytoplasmic membrane without

disrupting membrane function[33]. FM 4-64 is broadly applicable across bacterial species and compatible with live-cell imaging[66–77]. However, our segmentation model also works with other labels—including the membrane-permeable MitoTracker Green and with GFP fused to a membrane protein (Fig. 2d; Supplementary Fig. 4)—and, in principle, with any dye that provides a uniform signal along the cell periphery. In addition, we provide a transformation model, FP2FM, that predicts membrane-like boundaries suitable for segmentation from cytoplasmic fluorescence, adding flexibility (Supplementary Fig. 6). Altogether, the available dyes, combined with the transformation tools provided here and elsewhere[15], extend the applicability of the MEDUSSA method to a broad range of species. For direct comparisons, we recommend using the same staining modality and image transformation procedures, as different labels or transformation models can slightly affect size estimates. For example, FM 4-64 labels the outer leaflet of the cytoplasmic membrane, whereas GFP fused to a membrane protein is displaced toward the cytoplasmic side. Consequently, boundaries inferred from GFP-based signals may appear slightly inward relative to those inferred from FM 4-64, leading to systematic differences in estimated cell size (Supplementary Fig. 12).

Although MEDUSSA addresses several limitations associated with cell-size estimation from phase-contrast images, it does not overcome challenges inherent to the imaging of small bacterial cells by optical microscopy. A single pixel in a fluorescence microscopy image may amount to 10% of the width of a typical bacterial cell, so even minor segmentation imprecisions can translate into appreciable differences in measured size. For reliable comparisons between strains and replicates, it is therefore important to quantify sufficiently large numbers of cells and compare the size distributions of the populations. Bacterial cell size also depends on growth conditions[78], and biological variation between cultures can be substantial depending on media and growth stage[79–81]. We therefore recommend keeping growth conditions as constant as possible to obtain biologically meaningful and reproducible results.

Segmentation accuracy also depends on image quality. In particular, images with low signal-to-noise ratios can result in lower segmentation accuracy (Supplementary Fig. 13). To mitigate this limitation, we trained an additional CARE restoration model (FM2FM-HiSNR; see methods for details) that predicts high signal-to-noise deconvolved membranes across a range of input signal-to-noise levels. Under our imaging conditions, restoration with this model improves the segmentation quality of low signal-to-noise images (Supplementary Fig. 13), although extremely low signal levels will ultimately limit segmentation accuracy.

In addition, projection artifacts can lead to cell-size misestimation from fluorescent images, an effect that may be particularly pronounced for smaller cells[4,37]. Hardo et al. reported marker-dependent biases, with membrane fluorescence tending to underestimate cell width and cytoplasmic fluorescence tending to overestimate it, and with proportionally larger effects in thinner cells (below ~1 μm) than in thicker cells. To probe for this effect under our imaging and analysis conditions, we compared cell width estimates derived from independently segmented membrane and cytoplasmic fluorescence across a controlled width range (*ponA* under IPTG; Supplementary Fig. 14). If projection artifacts were prominent under our conditions, we would expect cytoplasmic-fluorescence-based estimates to be relatively larger than membrane-based estimates for thin cells, with the discrepancy decreasing as cells become thicker. However, width scaled almost identically when estimated from membrane and cytoplasmic

## a Representative micrographs of strains

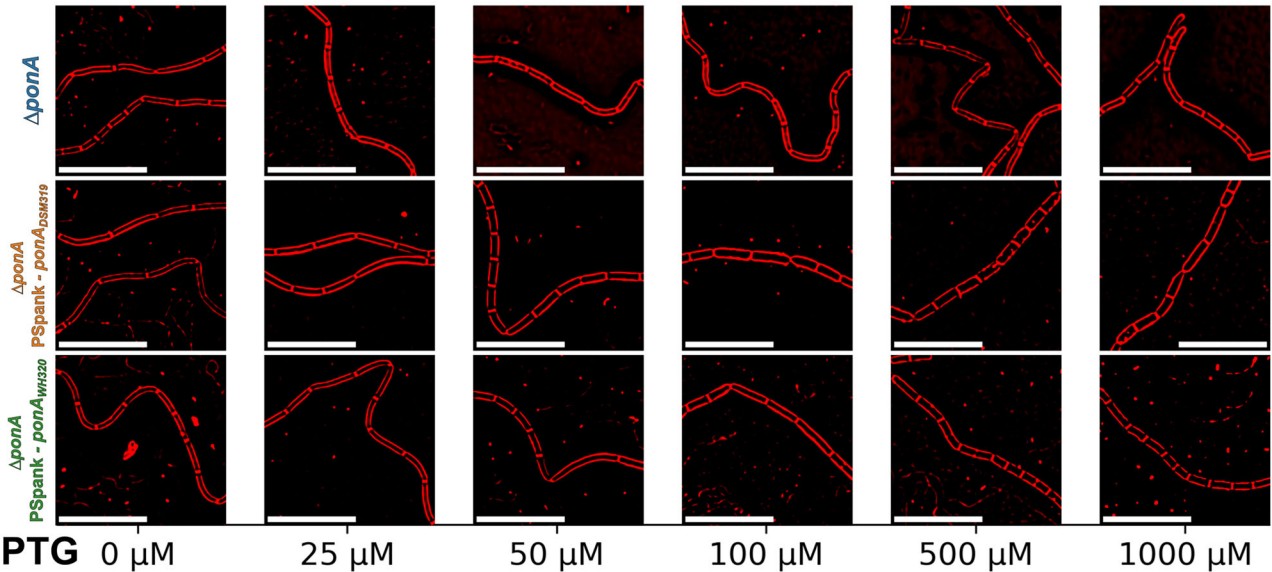

## b Inductions of *ponA* in *B. subtilis*

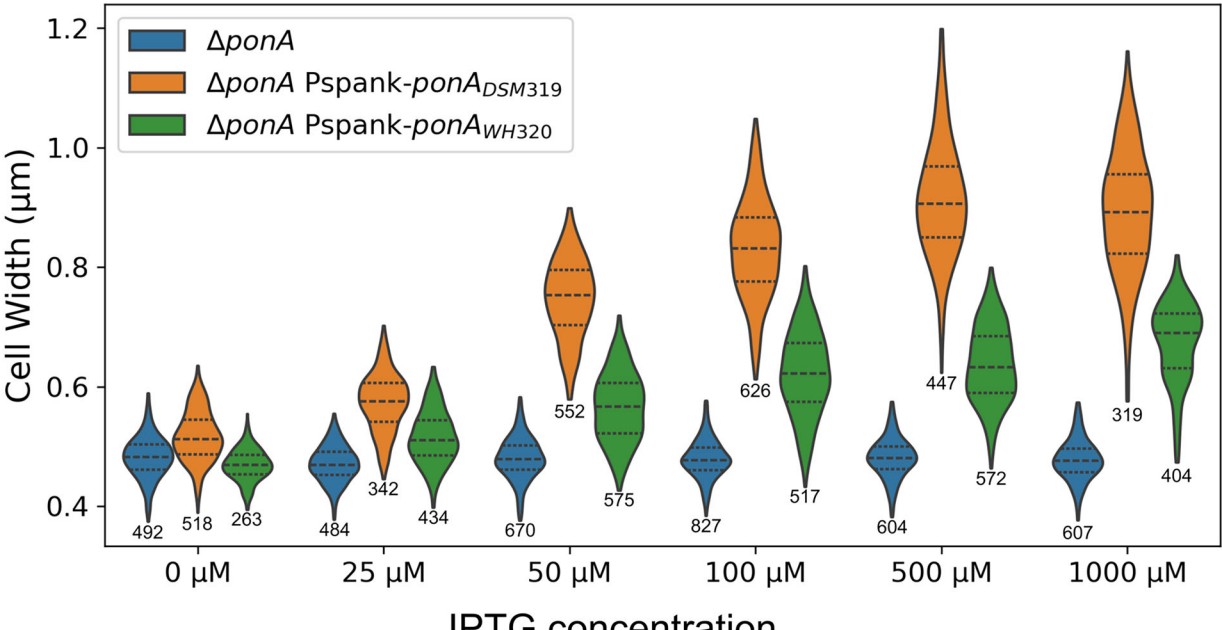

**Fig. 6 | WH320 carries a hypomorphic PBP1 allele. a** Representative FM 4-64-stained cells corresponding to panel b. Scale bar, 10 µm. **b** Cell width of a *B. subtilis* PY79 *ponA* mutant (Δ*ponA*, blue) and the same mutant expressing *P. megaterium* DSM 319 *ponA* (Δ*ponA* P$_{spank}$-*ponA*$_{DSM319}$, orange) or *P. megaterium* WH320 *ponA* (Δ*ponA* P$_{spank}$-*ponA*$_{WH320}$, green) from the IPTG-inducible P$_{spank}$ promoter inserted at the *amyE* locus. Violin plots show width distributions at different IPTG concentrations. Central 99% of data points was sampled for each strain in each condition. The dashed line indicates the median; dotted lines mark the first and third quartiles. The number of cells for each violin is shown in the plot beneath it. Prior to measurement, segmentation output was manually curated to remove aberrant masks and truncated masks at image edges. Width distributions of the three strains differed significantly at all IPTG concentrations (Kruskal test).

fluorescence across the tested width range (~0.6–1.2 µm) (Supplementary Fig. 14b). These results suggest that projection artifacts, as described by Hardo et al., do not measurably bias our width estimates under these conditions, although they may become more relevant in cells thinner than those examined here or under different imaging settings.

In this study, we applied MEDUSSA to quantify the size of six strains of *P. megaterium*. Our measurements reveal remarkable diversity in cell size both among strains and among individual cells within a strain (Fig. 5). Although all strains produce comparatively large cells by bacterial

standards, some produce cells that are, on average, over twofold larger in volume than others. We also observed that some strains formed extremely elongated cells during exponential growth, which were not observed in the stationary phase. To our knowledge, filamentation has not been reported previously in *P. megaterium*—likely because membrane dyes are not routinely used in microscopy. The mechanism underlying this filamentation, and whether such filaments occur in nature or are artifacts of laboratory culture, remains to be determined.

Cell width in bacteria is generally assumed to be tightly controlled within a species, with little cell-to-cell variation in steady state cultures[54,55,57,82–84]. Our dataset, however, reveals substantial width variation both within and between strains. Within strains, width values were approximately normally distributed, and some strains showed wide cell-to-cell variation; for example, in strain KM during stationary phase, the thinnest cells were up to twofold thinner than the thickest cells (Fig. 5). The coefficient of variation for width reached up to 0.15 (Supplementary Fig. 11) —more than twice that of *B. subtilis*—suggesting that mechanisms of width homeostasis in this bacterium are less precise than in *B. subtilis*[53,54].

Between strains, width distributions were shifted, with median widths ranging from ~0.9 to 1.4 µm (Fig. 5; Supplementary Fig. 11a), indicating variation in width regulation across strains. We further examined differences between the thick strain DSM 319 and its Lac⁻ derivative WH320, which is ~0.5 µm thinner. Genome comparison revealed an alanine-to-aspartic acid substitution in the transglycosylase domain of PBP1. An alignment of PBP1 sequences from 854 strains indicates that alanine at position 214 is largely conserved (~75%); ~24% carry serine at this position, and fewer than 1% contain other residues (e.g., cysteine, threonine, valine) (Supplementary Data in Zenodo repository). Notably, no sequence harbors an aspartic acid or any other charged residue at this position, suggesting that introducing a charge there impairs PBP1 activity. Because PBP1 mutants yield thinner cells in *B. subtilis*[45], it is plausible that the hypomorphic PBP1 allele in *P. megaterium* WH320 contributes—at least in part—to its reduced width relative to the DSM 319 parental strain.

Unlike the mutation in *lacZ*, the *ponA* mutation in *P. megaterium* WH320 is a G→T transversion. Because EMS typically induces G·C→A·T transitions[58], it is unlikely that the *ponA* mutation arose directly from EMS mutagenesis. Instead, we suspect that the *ponA* mutation in strain WH320 was either already present in the DSM 319 clone used for EMS treatment or was acquired during subsequent propagation of WH320 before we received the strain.

Overall, these results illustrate how MEDUSSA can be used to explore within- and between-strain diversity in bacterial cell size and to reveal differences that can be pursued to identify underlying mechanisms. Future work will expand our strain set and survey additional species and environmental isolates to advance our understanding of the evolution of bacterial cell size and contribute to the emerging field of evolutionary cell biology[85,86].

## Methods
### Strains and plasmids
A list of strains is provided in Supplementary Table 1. Plasmid and primer lists are in Supplementary Tables 2 and 3, along with detailed descriptions of plasmid construction.

### Culture media and growth conditions
Bacteria were routinely grown on LB agar plates (Lennox formulation, 5 g NaCl/L) overnight at 30 °C, with the exception of *Lactococcus lactis*, which was grown on MRS medium at 28 °C. *Saccharomyces cerevisiae* was grown on YPD at 37 °C.

For size quantification of *B. subtilis* and *P. megaterium* strains, a single colony was inoculated into 5 mL fresh LB Lennox and grown overnight in a roller drum at 30 °C with constant rotation. The next morning, cultures were diluted 1:1000 into 30 mL of the appropriate medium in 250-mL baffled flasks and incubated in a water bath at 30 °C with shaking at 220 rpm. Samples were taken after five hours (exponential; Supplementary Fig. 10) and after ten hours (stationary; Supplementary Fig. 10) for microscopy (see next section).

For the induction experiments shown in Fig. 6 and Supplementary Fig. 14, a single colony was inoculated into LB Lennox diluted 1/4, then grown at 25 °C in a shaking incubator at 220 rpm. The next morning, cultures were diluted to an OD600 of 0.02 in 1/4-diluted LB Lennox with the indicated IPTG concentration (0, 25, 50, 100, 500, or 1000 µM). Cultures were grown at 30 °C in a shaking incubator to an OD600 of 0.2–0.5 and then samples were taken for microscopy.

Sporulation (Fig. 4e) was induced by resuspension, as described elsewhere[87].

### Microscopy
For imaging, 10 µL culture samples were imaged on 1.2% agarose pads in 1/4-diluted LB-Lennox supplemented with 0.5 µg/mL FM4-64. The pad was covered with a #1.5 glass coverslip. Ten-hour samples were diluted three times to avoid overcrowded fields.

For cell-size estimation experiments for the different *B. subtilis* and *P. megaterium* strains, cells were imaged using a Leica Thunder Imager live cell with a Leica DFC9000 GTC sCMOS camera and a 100X HC PL APO objective (Numerical aperture 1.4, Refractive index 1.518 objective). FM 4-64 images were taken using a 510 nm excitation LED light source, with 70% intensity and 0.1 s exposure, and a custom filter cube produced by Chroma Technology GmbH (Olching, Germany) with the following components: excitation filter, ET505/20x; dichroic mirror, T660lpxr; emission filter, ET700lp. Training images for the restoration (FM2FM and FM2FM-HiSNR) and translation (FP2FM) models were taken with a DeltaVision Ultra microscope with a PCO Edge sCMOS camera and an Olympus 100X UPLX APO objective (Numerical aperture 1.45, Refractive index 1.518 Ph3). FM 4-64 and mVenusQ69M were visualized with the following filters: excitation 542 nm and emission 679 nm for FM 4-64, and excitation 475 nm and emission 525 nm for mVenusQ69M, with light transmission at 100% and an exposure time of 0.2 s for both channels. Training images for the segmentation models (FMSeg, RawFMSeg) were taken using the same illumination conditions as the restoration model. For the membrane images of *E. coli*, the FM4-64 was captured with light transmission of 40% and an exposure time of 0.1 s.

For phase-contrast microscopy, the aforementioned DeltaVision microscope was used, with light transmission set to 100% and exposure time to 0.1s.

For microscopy comparisons, we also used a Zeiss Axio Imager Z2 microscope with a Hamamatsu sCMOS camera and a Zeiss 100x Plan-Apochromat objective (Numerical aperture 1.40, Refractive index 1.518 Ph3). FM 4-64 was visualized with an excitation filter of 506 nm and emission filter of 751 nm, with light transmission at 100% and an exposure time of 0.2 s.

### Ground truth image annotation using active contours
We annotated cells in our ground-truth images using the ImageJ plugin JFilament[36], which allows the semi-automatic generation of sub-pixel resolution active contours (snakes) that follow the ridges of maximum fluorescence membrane intensity around the cell (Fig. 1f; Supplementary Fig. 1)[3,71,88]. Default parameters were used, except for Gamma, which we increased when the preset value led to excessive snake deformation. For cell clusters, snakes were typically manually corrected because contour edges tended to deform toward neighboring cells. When membrane blobs distorted cell contours, we also corrected the snakes to follow straight lines. These steps ensured consistent single-cell segmentation during model training and improved the segmentation model's robustness to such confounding artifacts.

### Training and evaluation of models for segmenting fluorescent membranes
We annotated 143 raw and SoftWorx (Cytiva Life Sciences)-deconvolved FM 4-64 membrane images using JFilament[36], as described above, yielding over 7000 cells for training. Training images were maximum-intensity projections of FM4-64 z-stacks. Following the Omnipose documentation[14], we trained the models from scratch, i.e., not using any existing model parameters as starting points. In contrast, for Cellpose3[35], Cellpose-SAM[21] and microSAM[20], the existing base models were fine-tuned with the existing data. All trainings were done following the suggestions found in their respective documentation. Omnipose models were trained with a batch size of 8 images for 4000 epochs using a learning rate of 0.1 and a Rectified Adam (RAdam) optimizer[89]. Both models (raw and deconvolved) were trained

using the same parameters. Cellpose3 models were trained with a batch size of 1 image for 100 epochs using a learning rate of 0.1, a weight decay of $1\times10^{-4}$, and a Stochastic Gradient Descent (SGD) optimizer[90]. Both models (raw and deconvolved) were trained using the same parameters. Cellpose-SAM models were trained with a batch size of 1 image for 100 epochs using a learning rate of $1\times10^{-5}$, a weight decay of 0.1, a tile size of 256×256 px, and an AdamW optimizer[91]. Both models (raw and deconvolved) were trained using the same parameters. MicroSAM models were trained with a batch size of 1 image for 10 epochs of 1788 iterations per epoch, using a learning rate of $1\times10^{-5}$, a weight decay of 0.1, a tile size of 512×512 px, and an AdamW optimizer. Segmentation decoder was also trained to allow for automatic instance segmentation. Both models (raw and deconvolved) were trained using the same parameters.

Training was done in the "deepbio" computing cluster at the Max Planck Institute for Evolutionary Biology, using a single NVIDIA(R) GeForce GTX 2080 Ti GPU with 12GB memory.

For accuracy assessment, we annotated 56 images with JFilament. F1 score[92–94] was calculated at eleven Intersection over Union (IoU) thresholds spanning 0.5 to 1, in 0.05 intervals. True positives, false positives, and false negatives were calculated using the Omnipose library function "average_precision". F1 was then computed using these values following Caicedo et al. 2019[92].

### Simulation of fluorescent membranes

Synthetic rod-shaped cells were generated using SyMBac[95], with a constant length of 4 μm and variable widths, at a pixel size of 15 nm. Contours were extracted from each plane of each synthetic cell to represent the membrane. To simulate membrane staining, 90% of contour points were randomly selected for the placement of fluorescence molecules. To model light spread, a synthetic point spread function was generated in SyMBac using the "3 d fluo" mode with the following parameters: radius = 150, wavelength = 600 nm, numerical aperture = 1.49, refractive index = 1.5, and pixel size = 15 nm. To measure cell width from JFilament-generated masks (Supplementary Fig. 1), simulated images were rescaled to a pixel size of 65 nm to match our microscopy images. Measurements were performed on the middle focal plane of the simulated z-stacks.

### Training and evaluation of models for deconvolution prediction (FM2FM) and fluorescence translation (FP2FM)

Deconvolution-prediction and fluorescence-translation models were trained using the Content Aware Restoration (CARE) framework[40]. For FM2FM and FP2FM model, 200 images (2048×2048 pixels) of vegetatively growing *B. subtilis* constitutively expressing mVenusQ69M[96] were acquired using a DeltaVision Ultra microscope with a PCO Edge sCMOS camera, capturing FM4-64 membrane fluorescence, and mVenusQ69M cytoplasmic fluorescence. Images were deconvolved in the DeltaVision software (SoftWorx) and then cropped to remove deconvolution edge artifacts, yielding 1914×1914 px image outputs. Non-deconvolved training inputs were cropped to the same size. Training input images were single-plane images of raw FM4-64 (for the FM2FM model) or mVenusQ69M (for the FP2FM model) z-stacks, and target images were the deconvolved FM4-64 images for both models. For training, 16 patches of 512×512 px were extracted from each image pair. Models were then trained using a batch size of 32 images for 100 epochs, with 100 training steps per epoch. Training was performed in the "deepbio" computing cluster at the Max Planck Institute for Evolutionary Biology, as above. Model parameters and training hardware was the same for both models.

To compare model performance against the membrane deconvolution target (SoftWorx), we conducted qualitative assessments of output images' intensity profiles. We quantified the Structural Similarity Index Measure (SSIM) between the extracted image patches. Patches were selected to maximize the number of cells and minimize the background. SSIM was calculated using the Python scikit-image library[97]. We then statistically compared cell-size estimates using membrane images deconvolved with SoftWorx or predicted with FM2FM or FP2FM (Fig. 3; Supplementary

Fig. 6). Width, Length, Surface Area, and Volume were measured using the skeletons as explained further down. Cross-sectional area, convex hull area, eccentricity, and solidity, were directly obtained from segmentation masks using the Python scikit-image library[97].

### Training and evaluation of a deconvolution-prediction model that increases the signal-to-noise ratio (SNR) of predicted images (FM2FM-HiSNR)

We use CARE[40] to train an additional deconvolution prediction model that produces high signal-to-noise ratio (SNR) images from low-SNR non-deconvolved fluorescent membrane images. We named the model FM2FM-HiSNR. FM 4-64 images of vegetatively growing *B. subtilis* were acquired at four different illumination conditions using a DeltaVision Ultra microscope with a PCO Edge sCMOS camera. The illumination conditions were as follows (exposure time and percentage of transmitted light listed after their names in quotation marks): "Noisy", 0.01 s, 10%; "i1", 0.01 s, 50%; "i2", 0.025 s, 10%; "Target", 0.2 s, 100%. Images were deconvolved using the DeltaVision software (SoftWorx) and then cropped to remove deconvolution edge artifacts, yielding 1914×1914 px image outputs. Non-deconvolved training inputs were cropped to the same size. Training input images were single-plane images of raw FM4-64 z-stacks from all different illumination conditions, and target images were the deconvolved FM4-64 images at the "Target" illumination condition. In total, 98 input images were used. For training, 16 patches of 512×512 px were extracted from each image pair. The model was then trained using a batch size of 32 images for 100 epochs, with 100 training steps per epoch. Training was performed in the "deepbio" computing cluster at the Max Planck Institute for Evolutionary Biology, as above.

To explore the relationship between SNR and segmentation accuracy, and to test whether restoration with FM2FM-HiSNR improve segmentation accuracy of low SNR images, FM 4-64 images of vegetatively growing *B. subtilis* were acquired at 12 different illumination conditions using a DeltaVision Ultra microscope, under the following illumination conditions (exposure time and percentage of transmitted light listed after their names in quotation marks): "Noisy", 0.01 s, 10%; "i1", 0.01 s, 50%; "i2", 0.025 s, 10%; "i3", 0.025 s, 50%; "i4", 0.05 s, 10%; "i5", 0.05 s, 50%; "i6", 0.1 s, 10%; "i7", 0.1 s, 50%; "i8", 0.1 s, 100%; "i9", 0.2 s, 10%; "i10", 0.2 s, 50%; "Target", 0.2 s, 100%. Images were deconvolved using the DeltaVision software (SoftWorx) or restored with FM2FM-HiSNR, and segmentation accuracy was calculated (Supplementary Fig. 13).

### Extracting cell size information from masks of rod-shaped cells

For rod-shaped bacteria, we assumed a sphero-cylindrical geometry with hemispherical caps[56]. First, we computed the minimum Euclidean distance from each pixel inside the mask to the closest pixel on the mask boundary[98], which is commonly referred to as the distance transform. Next, we generated a one-dimensional centerline (commonly known as "skeleton")[99], overlaid it on the distance transform map, and determined the distance value at each skeleton pixel, thereby obtaining the local radius of the mask along the skeleton line (Fig. 4a). An alternative one-dimensional representation is the medial axis[100], also known as the "topological skeleton" (Supplementary Fig. 15). We compared size estimates obtained with the two representations and found significant differences in some size metrics in a set of *E. coli* cells (Supplementary Fig. 15d). Inspection of individual masks revealed that some medial-axis representations tended to over-branch (Supplementary Fig. 15e). If common across many cells, such over-branching would lead to inaccurate size estimates. We therefore proceeded with the skeleton representation.

To estimate the extent of the hemispherical caps, we took the distance between each skeleton endpoint and the corresponding mask tip as the cell radius at the skeleton endpoints. This yielded precise estimates of mask length, and with profiles at every pixel along the skeleton. Assuming rotational symmetry and that the masks derive from medial focal-plane images, we estimated cell volume and surface area by adding the volumes and lateral surface areas of cylindrical segments generated by circularly projecting the

radius of the mask at each pixel of the skeleton, plus the hemispherical caps (Fig. 4a). This approach accurately handles elongated masks, provides local morphological information of cell morphology, and is robust to cellular curvature. While tailored to rods, the framework can be adapted to other morphologies.

**Dealing with branching skeletons.** Due to the mask geometry, it is possible that the skeleton branches into multiple endpoints (Supplementary Fig. 15c). To handle both branched and non-branched cases, we developed the following workflow. First, we treat the skeleton as a graph, with each pixel considered a node. Assuming connectivity only between neighboring pixels (horizontal, vertical, or diagonal), we define edges for pairs of pixels whose Euclidean distance is less than 1.5 pixels, thereby retaining connected neighbors. Next, we implement a Depth-first search (DFS) algorithm to find the longest possible path connecting the opposite endpoints in the graph. If the skeleton is not branched, this path corresponds to the skeleton. For branched skeletons, multiple candidate paths exist. For each candidate path, we compute the cell size metrics described in the next section and take the median of each metric as the cell size (Supplementary Fig. 15).

**Cell size measurements.** We combined the mask skeleton and distance transform to derive cell measurements under a spherocylindrical geometry with hemispherical caps and rotational symmetry. The measurements are implemented in a custom code written in Python3, which can be found in the GitHub repository linked in the "Code availability" section.

Cell length ($L$) is the sum of Euclidean distances $d$ between two successive points in the mask skeleton, plus the distance from each skeleton endpoint and the corresponding mask tip, which corresponds to the local radius at the skeleton endpoint $r_s$. With that, $L$ follows the equation:

$$L = r_{s1} + r_{s2} + \sum_{i=0}^{n-1} d(i, i+1) \tag{1}$$

Cell width ($w_i$) is calculated as twice the value of the radius at each point in the skeleton ($r_c$). Mean $w$ can then be calculated as:

$$\bar{w} = \frac{2}{n} \sum_{i=0}^{n} r_{ci} \tag{2}$$

Cell surface area ($S$) is calculated as the sum of lateral surface areas of the cylinders projected at each point in the skeleton, with radius $r_c$ and height $h$ of 1 pixel, plus the lateral surface area of the hemispherical caps with radius $r_s$. Thus, $S$ is calculated as:

$$S = 2\pi \left( r_{s1}^2 + r_{s2}^2 + h \sum_{i=0}^{n} r_{ci} \right) \tag{3}$$

Cell volume ($V$) is obtained similarly to $S$, as it adds the volume of each cylinder and the hemispherical caps (with same parameters of radius and height as $S$). $V$ is then obtained from:

$$V = \pi \left( \frac{2}{3} r_{s1}^3 + \frac{2}{3} r_{s2}^3 + h \sum_{i=0}^{n} r_{ci}^2 \right) \tag{4}$$

**Measurement transformation**
We segmented 864 single cells of *Escherichia coli* stained with FM 4-64 using either JFilament (ground-truth) or FMseg. Cells were from early exponential phase ($OD_{600} = 0.1$-$0.3$) cultures in either LB Lennox, or MOPS minimal medium with 0.1% glucose, providing a range of cell widths from 0.53 to 1.04 μm, and a range of cell lengths from 0.87 to 9.61 μm (based on measurements from ground-truth masks) (Supplementary Fig 8a). Measurements from ground-truth masks were plotted against the values

obtained from on the FMseg-generated masks, revealing a consistent size overestimation for FMseg masks (Supplementary Fig. 8b). The data show a linear relationship for the overestimation, which allows us to leverage statistical tools to compute the necessary parameters to transform the data between the automatic segmentation results and the ground truth results. A linear regression would be a direct way to do this, extracting the values for the slope ($m$) and intercept ($n$) necessary to transform the data. While linear regression returns a coefficient of error that would allow us to simulate distributions to sample $m$ and $n$, there is the chance that we may over- or under-estimate these values, as well as sample pairs that do not make sense in the context of our data. Thus, we leveraged the PYMC library[101] and its use of Bayesian statistics for sampling the necessary parameters, which are $m$, the fit slope, $n$, the fit intercept, and $\sigma$, the standard deviation of our sampled observations. Since our data behaves linearly, we used a Bayesian Generalized Linear Model (GLM), assuming that our parameters (slope, intercept, size metric, and its standard deviation) follow certain types of distributions. For $\sigma$, we assume a Half-Cauchy distribution, which only returns positive, non-zero values, which is the only type of values a standard deviation can have. For $m$ and $n$, we sample from normal distributions with mean 0 and standard deviation 20, giving us a wide range of possible parameters. Then, for computing the observed values, we also assume a normal distribution. PYMC performs a series of draws to fit the data and thus after successive draws we end with the distributions of $m$, $n$ and $\sigma$. In our case, we performed four chains of 3000 draws each, giving us a total of 12000 sampled values. We can then randomly sample pairs from these distributions in order to transform the data, as well as to include the uncertainty of the transformation in the analysis (Supplementary Fig. 8b). For the examples shown in Fig. 5, Fig. 6, and Supplementary Fig. 11, we sampled 250 pairs of m and n for each individual cell measurement, thus allowing us to compute new metrics.

**Statistics and reproducibility**
For the statistical tests in Fig. 3, Fig. 6, Supplementary Fig. 6, and Supplementary Fig. 15, Levene's test was computed for the distributions of interest. If the p-value was greater than 0.05, one-way ANOVA was performed to obtain the p-values and determine if the difference between distributions was significant or not. In cases where Levene's test was significant ($p < 0.05$), Kruskal test was used instead. All three tests were used as they are implemented in SciPy[102]. Each individual cell was considered as a replicate for the purposes of the statistical tests. Sample sizes are specified either in the figure legends or inset in the plots.

**CryoEM image analysis**
Because the high magnification of the cryo-EM images typically did not capture entire cells, we estimated cell width as the straight-line distance between the mother-cell lateral membranes, measured perpendicular to the cell longitudinal axis. In cryo-FIB tomograms of wild-type cells, only a portion of the cellular volume was retained: focused ion-beam milling produced a thin lamella that sampled a section of the cell. We analyzed only tomograms in which this section included the mid-cell region. To verify this, we measured width across successive z-slices and selected tomograms showing a clear width peak at mid-cell that decreased in slices above and below.

**Whole genome sequencing**
For whole-genome sequencing, *Priestia megaterium* WH320 was streaked in an LB-Lennox agar plate and grown overnight at 30°C. Next morning, a single colony was picked and re-streaked in an LB-Lennox agar plate and grown overnight at 30°C. Cells were collected from the plate and genomic DNA was extracted with the MasterPure Gram Positive DNA Purification Kit (Biosearch technologies). Extracted genome was then sequenced using an Illumina NextSeq 500 sequencer. Raw pair-end sequences were loaded into Geneious prime version 2025.1.2 for sequence trimming and pairing with an insert size of 500 base pairs. Paired sequence list was then exported in fastq format for analysis using breseq[103].

## Software and data availability

The fine-tuned segmentation models for raw and deconvolved images (for Omnipose, Cellpose-SAM, microSAM, and Cellpose3), deconvolution prediction (FM2FM), deconvolution prediction and SNR increase (FM2FM-HiSNR2), and fluorescence translation (FP2FM) models can be found in the Zenodo (https://zenodo.org/records/18978187). Segmentation model training and benchmarking data can be found at the BioImage Archive repository https://www.ebi.ac.uk/biostudies/studies/S-BIAD2350. Deconvolution model training and benchmarking data can be found at the BioImage Archive repository https://www.ebi.ac.uk/biostudies/studies/S-BIAD2353. Images of individual *E. coli* cells used for measurement transformation with their corresponding annotations and automatic segmentations be found at the BioImage Archive repository https://www.ebi.ac.uk/biostudies/studies/S-BIAD2352. Segmentation of *B. subtilis* and *P. megaterium* strains (wild-type and induction constructs) be found at the BioImage Archive repository https://www.ebi.ac.uk/biostudies/studies/S-BIAD2354. *P. megaterium* WH320 genome sequence has been deposited in ENA (accession number PRJEB112284). PBP 1 A/1B protein sequence alignment, and results from breseq[103] are available in Zenodo[104]. Raw data files for reproducing the plots are available in Zenodo[105]. All other data are available from the corresponding author on reasonable request.

## Code availability

The MEDUSSA GitHub repository contains all software-related material (https://github.com/OReyesMatte/MEDUSSA). The code version used to produce the manuscript figures, as well as the raw data, is available in Zenodo[105]. The "MEDUSSA" folder contains the functions for size measurement (measure.py), Bayesian sampling, and measurement transformation (both in transform.py). The "CARE" folder contains the environment file which was used for training the deconvolution models, as well as the corresponding notebooks for preparing the data, train the model, and test it. The "Segmentation" folder contains the environment files for the different segmentation frameworks, as well as examples for both training commands and uploading and running the models. The "Figures" folder contains notebooks to reproduce the plots shown in the manuscript. The "Raw data" folder contains all data used to make the manuscript plots.

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

## Acknowledgements

We are grateful to Elitza Tocheva, Elizabeth Villa and Kanika Khanna for kindly sharing CryoEM images of *B. subtilis* sporulating cells, Kaumudi H. Prabhakara for providing the *Saccharomyces cerevisiae* and *Lactococcus lactis* strains used in Fig. 2d, and Felipe Sinhorini for helping with the acquisition of the BGSC strains. We thank members of the Evolutionary Cell Biology group, the Ojkic lab, and the Microbial Population Biology department for continuous discussion and feedback. We thank Elisa Brambilla and Michael Schwarz for helping with the acquisition of images with the Zeiss Microscope. We also thank the members of the Scientific IT group at the Max Planck Institute for Evolutionary Biology for computational support.

## Author contributions

Conceptualization, O.R.M., N.O., and J.L.G.; Methodology, O.R.M. and J.L.G.; Software, O.R.M., C.F.G., and B.G.; Validation, O.R.M.; Formal analysis, O.R.M., C.F.G., and B.G.; Investigation, O.R.M. and N.H.; Resources, N.H.; Data Curation, O.R.M.; Writing - Original Draft, J.L.G.; Writing - Review & Editing, All authors; Visualization, O.R.M., N.O., and J.L.G.; Supervision, N.O. and J.L.G.; Project administration, O.R.M. and J.L.G.; Funding acquisition, J.L.G.

## Funding

JLG discloses support for publication of this work from Funder ERC starting grant 853323, and the Max Planck Gesellschaft. NO discloses support for publication of this work from Funder BBSRC grant BB/Y009002/1. Open Access funding enabled and organized by Projekt DEAL.

## Competing interests

The authors declare no competing interests.
