## [Transparent Peer Review file · Communications Biology]

Deep-learning deconvolution and segmentation of fluorescent membranes for high-precision bacterial cell-size profiling

Corresponding Author: Dr Javier Lopez-Garrido

Version 0:

Reviewer comments:

Reviewer #1

(Remarks to the Author)

In their manuscript, Reyes-Matte et al. present a framework for deep learning (DL) based cell segmentation and downstream analysis of bacterial morphology. The proposed method is based on the Omnipose network and, according to the authors, provides robust cell segmentation across various cell shapes using standard membrane staining protocols. One problem of phase contrast microscopy is the inability to segment filamentous bacteria that consist of several individual cells not discernible in phase contrast images. The authors tested various “out-of-the-box” DL models dedicated to segment bacterial bioimages. The poor performance motivated them to develop their image analysis pipeline MEDUSSA. This pipeline used DL-based segmentation of deconvolved membrane-stained images and was used to compare cellular dimensions of various *Bacillus* and *Priestria* strains across different growth stages. Morphological analysis revealed an unexpected phenotype that the authors traced back to a point mutation in the PBP1 protein. Complementation assays verified that PBP1 expression increases cell diameter, while inactivation or deletion leads to the observed narrow-cell phenotype. In addition to the segmentation model, the MEDUSSA framework also includes two image restoration models that can be used to predict deconvolved images and infer a membrane stain from cytosolic labeling. The README file in the Github repository provides information about installation of the required packages and some information about model functions and parameters. Actual guidelines on how to use MEDUSSA properly on custom data are not provided, though. What we appreciate a lot is that code and data are shared publicly, making it a valuable resource for other researchers.

While the manuscript is well-written and data analysis was carried out carefully, the impact and relevance of the presented work is not evident. The performance of the MEDUSSA pipeline is not compared to new DL networks, in particular foundation models (e.g. CellposeSAM). Furthermore, it is not clear whether the presented pipeline extracts correct information, or whether the results are systematically affected by projection artifacts. Quantification of model performance could be improved, in particular for the image restoration networks. Finally, the results are not discussed critically but seem to be presented overly optimistic.

In the following, we will elaborate on major/minor issues and hope that our comments are helpful to the authors.

Major points:

- 1) CellposeSAM (as of May 2025) represents a significant advance over previous cell segmentation methods by robustly generalizing across imaging modalities, data qualities, and cell sizes, now outperforming both inter-human agreement and previous benchmarks for accuracy in most common conditions. Unlike classical U-Net-based models, its transformer backbone—adapted from the Segment Anything Model (SAM)—enables improved feature extraction and compositional generalization, supporting high precision segmentation even in challenging scenarios. Currently, there is no evidence that CellposeSAM systematically overestimates cell area in the same way that classical models sometimes do, especially on lower-quality or out-of-focus images.
- 2) The paper should highlight that CellposeSAM sets a new standard for microscopy segmentation, with accuracy surpassing inter-human agreement and notable robustness to image degradation and biological variability. It does not introduce an area overestimation bias as has been observed in previous models, provided training and data alignment are

performed appropriately.

3) What does MEDUSSA stand for? It seems to be an acronym, but it is not specified in the manuscript.

4) Lines 99-100: The authors state that they “trained a custom deep learning model to facilitate the segmentation of fluorescent-membrane images”. It is, however, not mentioned that this model is just a custom-trained Omnipose model instead of an extension or modification of a NN architecture. This should be mentioned in the Abstract and also throughout the manuscript, not only in the methods section. Training a published DL architecture on custom data is not a big deal nowadays and raises the question of whether this work is a significant contribution to the field.

5) Figure S1: The comparison of selected segmentation approaches on membrane images of different bacterial species is useful. However, it is not clear whether optimal parameters were chosen for these models. For example, Omnipose requires rescaling of the data so that the cell width is in a suitable range. Has this been taken into consideration? Also, there are other methods that were shown to work on diverse morphologies and imaging modalities (e.g. Ursell et al., Morphometrics). Is there a reason why the authors did not test these models on the test datasets? We are aware of the focus on Deep Learning based methods, but if methods based on conventional algorithms or classical machine learning perform well, we would prefer those over DL methods due to better explainability. 6) Lines 162 – 170: What is the benefit of the image restoration model? It is expected that they perform similar to the method used to generate the ground truth (here deconvolution). Is the CARE prediction faster? If not, why not use deconvolution instead of a DL model that requires rigorous training and testing? Regarding the FP2FM model, would it not be straightforward to perform segmentation on the cytosolic marker? The cells appear to be well discernible by the cytosolic signal, even when they grow in chains (Fig. 3a, bottom panel). Moreover, the performances of these models were not quantified. Can the authors validate via suitable metrics (e.g. SSIM or RSME) that the predictions are artifact-free and high-quality?

7) CARE can also be used to predict super-resolution images from widefield images (see DeepBacs, prediction of *E. coli* / *S. aureus* SIM images). Would such a network be more informative than predicting deconvolved images? As shown in Hardo et al. 2024, membrane stains in widefield imaging misestimate the underlying membrane position due to projection effects. This effect is actually shown in Fig. 4e, where the width determined in the deconvolved images systematically differs from the width of CryoEM data. Especially when it comes to volume comparisons, an underestimation of 15% at low cell widths can have a large effect on the volume ratio. This difference is not discussed in the manuscript, but instead the FMSeq segmentation is considered to be a “realistic cell size estimation” (line 260). However, this deviation is significant and should be discussed in detail.

8) Lines 239 to 249: In addition to the previous comment: The authors correct for a segmentation bias introduced by the model. However, it is not clear whether the ground truth annotation is actually accurate. Can the authors rule out that cell masks are underestimated during manual annotation, as suggested by Hardo et al., 2024? A benchmark with super-resolution or simulated data would be essential to validate that the “ground-truth” generated with FilamentJ can be considered as ground truth. This part is highly important, as a reliable cell size assessment can only be guaranteed by such benchmarks.

9) Line 282-283: The authors state that the results obtained by MEDUSSA are highly reproducible between replicates. However, the data shown in Supplementary Figure 6 show deviations of up to 15%. Is this deviation considered to be normal? The authors should discuss these deviations and also elaborate on whether they are of biological or analytical nature. Along these lines: Did the authors test different staining efficiencies? We assume that brightness of the membrane stain can affect the result, but this has not been assessed throughout this manuscript. Simulated data could be used to quantify the influence of different parameters (noise, brightness, ...) on segmentation quality and would allow comparison to the ground truth (see Hardo et al. 2024).

10) Hardo et al. show that the wavelength has a strong influence on the segmentation quality. Particularly fluorophores with emission at longer wavelengths increasingly underestimate the cell area. The authors should compare the segmentation quality of different fluorophores and investigate this effect in their data. Here, the data from the inner membrane GFP fusion (QoxB-sfGFP) could be compared to the FM4-64 stain.

11) Quantification of the performance of FMSeq on the species shown in Figure 2d would be of interest. Does the model perform equally well on the different datasets? Such quantification should be simple and rapid with the available code. It would also give partial answers to our previous comment.

12) Given our experience with DL models and particular CARE, the result of a CARE prediction is strongly dependent on the input image quality. Using data obtained under different optical configurations or microscope settings will eventually already result in artifactual reconstructions. This is even worse when applying a CARE model to data recorded on a different microscope. The authors should discuss how well the provided CARE model can generalize and showcase its performance on publicly available datasets of membrane-stained bacteria. This way it can be assessed whether the provided model can be used reliably or whether retraining is required when switching strains/settings/microscopes (we would expect the latter).

13) Figure 6: The results obtained from the complementation assays do not result in values comparable to the wild type strains. While we acknowledge that the observed trend matches the differences in cell width between the PBP1 variants of *Priestia megaterium*, we wonder what the reason for the discrepancy is (e.g. 0.9 μm width for WH320 vs. 0.7 μm in the fully induced complementation mutant). The authors should at least describe these deviations and provide possible explanations.

14) In general, the described PonA phenotype is rather dramatic with a width decrease of 35%. This difference does not require robust quantification as it is evident from brightfield images. Can the authors provide scenarios where subtle differences in cell dimensions need to be determined?

15) An overview on the image analysis pipeline of MEDUSSA would be useful. This could include the individual steps and alternative routes (e.g. whether and when CARE models are used for inference). Such a figure could be based on the overview provided in the README file of the Github repository. While reading the manuscript, it was actually not clear whether both models (FM2FM, FP2FM) are supported in the pipeline and whether you can choose to segment cells on non-deconvolved or deconvolved images.

16) The manuscript seems to be overly optimistic about the results obtained. Despite several deviations (15% difference between replicates; 15% difference in cell width between MEDUSSA and CryoEM results), there is no critical discussion included. The authors should provide an unbiased and honest view on the results and also discuss the limitations of their

algorithm. Particularly, the statement that MEDUSSA is applicable to virtually any bacterial species (line 409) is not justified given the small number of species analyzed in this work.

17) In general, the provided framework represents a collection of scripts and a combination of already published networks. This would hardly justify publishing the workflow as a novel approach, but rather represents a showcase of how to use a custom-trained Omnipose model with some downstream analysis. If there would be a dedicated software or plugin (napari, Fiji) together with a data post-processing module and a detailed documentation about proper usage, the framework would likely be more appealing to biologists that want to use the approach on their data.

Minor points:

1) Multi-label models were shown to provide more accurate segmentations than models that rely only on background and foreground (see work from David van Valen). Have the authors tested such models or tried to implement a multi-label approach into their model?

2) Introduction: The authors state that there are a limited number of studies regarding exact cell sizes and morphological heterogeneity of bacterial species. While this might be true for interspecies comparisons, there has been work on mutant libraries that also cover a wide range of morphologies. While such work is present as references in the manuscript, it is not mentioned in the introduction (e.g. Morphometrics <https://doi.org/10.1186/s12915-017-0348-8>) or BiofilmQ (<https://doi.org/10.1111/mmi.15064>). The authors should introduce methods that were already shown to work on a range of different phenotypes and samples.

3) Line 91: "lipophilic"

4) Lines 91/92: Different membrane stains exist that show altered performance on various species. Example is Nile Red that works much better on *Staphylococcus aureus* compared to the FM dyes. Other membrane markers exist, e.g. Mitotracker or fluorescent D-amino acids as cell wall precursors. The authors might want to extend their list of membrane markers beyond FM4-64. Reading further, alternative membrane labels were introduced in Lines 143/144). The authors might want to generalize labeling strategies already at this part of the manuscript.

5) Lines 129 and 132: Do the authors refer to Supplementary Figure 1 instead of 2?

6) Figure 4C: The blue line should be added to the plot legends for clarity

7) Fig. 5: Strain identifiers should be added to all panels for clarity.

8) Lines 310 – 315: The authors elaborate on rare events, such as extremely long cells. What is the frequency such events occur at? It is not clear whether such 'outliers' describe relevant biological phenotypes or just represent stressed cells that occur spontaneously.

9) Lines 415 – 417: Why should filamentation only be visible with membrane stains? Strongly elongated cells in brightfield or phase contrast images are usually considered to be filamentous, so the reason for this phenotype not being described for *Priestia megaterium* might be a lack of interest in this strain.

10) Line 425: The authors mention that a CV of 0.15 is untypically high. Can the authors include literature that supports this statement? What are typical CV values that are considered "normal" or represent a "tight regulation" of cell width?

11) Figure S4: Caption: 3000 thousand

Reviewer #2

(Remarks to the Author)

I co-reviewed this manuscript with one of the reviewers who provided the listed reports. This is part of the Communications Biology initiative to facilitate training in peer review and to provide appropriate recognition for Early Career Researchers who co-review manuscripts.

Reviewer #3

(Remarks to the Author)

The paper proposes a complete workflow to accurately segment bacteria from fluorescent membrane images, and to extract size parameters e.g. width, volume, area for comparative analysis across different bacterial strains. The core aim of the paper it appears is to set a standard workflow for morphometric analyses.

Generally, the paper presents the subject well, and motivates the need for fluorescent membrane imaging vs the current practice of using phase-contrast, in that the latter fails to delineate individual bacteria when they are compacted. Therefore, what appears to be one long bacteria is actually a collection of smaller bacteria. The authors also made efforts to benchmark each step of the pipeline and I applaud them for the detail in measuring the size discrepancy.

Couple of comments:

1. It is not so obvious to me the Omnipose architecture and more importantly the code implementation is the best in 2025. I would encourage authors to also retrain cellpose3, check and retrain cellpose-SAM, micro-SAM to see if these would not actually be better. In particular we see the out-of-box generalization performance of cyto3 in Fig.2a is miles better than omniopose.

2. The segmentation model appears to be trained on higher signal-to-noise ratio deconvolved images. How does the performance drop off with signal-to-noise ratio? with less optimal or high-resolution fluorescence imaging? is there a minimum required resolution?

3. The model performance appears to be evaluated with 16 images. Is this small number sufficient to get a stable

evaluation? How does the F1 score curve change as the number of images increases? It should presumably plateau. Did authors check this?

4. Authors claim the model trained on FM 4-64 membranes generalizes to other markers. However, there is no quantitative benchmarking to support this statement. What is the F1 score decrease?

5. Since authors explored training of models to predict deconvoluted images, couldn't you also train models to predict from phase-contrast images the FM 4-64 marker to bypass membrane marker choice?

6. model performance evaluation is confusing and somewhat unfair. e.g. in Fig.2a the retrained omnipose FMSeg is compared with off-the-shelf Cyto3 and bact_fluor_omni models. The fair comparison is to retrain all alternative models on the same dataset, to really show that omnipose architecture is indeed the best one. Also the pretrained models should be explicitly labeled as pretrained in the figure panel. The FM 4-64 is clearly an out-of-distribution dataset, and I don't expect foundation models to maximize their potential out-of-the-box.

7. How did authors generate the ground-truth cell widths? Is this from manual cell outlines? Would this not be subject to annotator bias, and quality of the fluorescence imaging? Presumably, the widths is sensitive to skeletonization, and ground-truth widths shouldn't use the skeletons, a derivative measurement, so did authors manually draw widths? It would be good to be specific and detailed about this which is the main motivation of the paper.

Reviewer #4

(Remarks to the Author)

In this work, Reyes-Matte et al. offer an insightful study of bacterial cell segmentation and the calculation of cell size from fluorescent images depicting cytoplasmic and cell-membrane-associated structures. The study is insightful because it showcases the sequential use of fluorescence microscopy, deconvolution via image-restoration methods (CARE), deep learning-driven image segmentation (Omnipose), and systematic error correction to extract physiologically relevant data, such as individual cell size. MEDUSSA provides an option to obtain accurate size measurements despite experimental heterogeneity in acquired images, such as differences in focal planes or morphological features not readily available in phase contrast microscopy, the most common microscopy regimen for bacterial segmentation. Although the study follows a clear logic and is well written, the authors should address the following points before publication:

1) Lines 129-132:

The authors state that the model "struggled with clustered cells, often producing indented or truncated masks (Supplementary Fig. S2)." This could happen because there were not enough images with clustered cells in the training data set, preventing the algorithm from generalizing across crowded conditions. What was the proportion of images with clustered cells in the training data set? Could the clustered segmentations have been improved by adding more clustered images to the training data set?

2) Lines 166-167:

To prove the following statement: "The fluorescence profiles of the predicted images closely matched those of deconvolved images (Fig. 3a,b), demonstrating the accuracy of the predicted deconvolution," the authors should provide a quantitative measurement of image similarity, for instance, the calculation of the structural similarity index SSIM for the area of the image with the membranes.

3) Lines 188-191:

The multiple size measurements are highly correlated parameters (length, area, volume, etc). A more comprehensive assessment could be performed using morphological descriptors that are not entirely associated with mask length, such as eccentricity, solidity, and convex hull.

4) Line 191:

The statement "The size distributions were nearly identical (Fig. 3c)" requires p-values or statistical tests to clearly assess all comparisons.

5) Line 201-202:

Provide a brief explanation of the algorithm in the text.

6) Lines 210 – 211:

“However, due to minor aberrations in microscopy slides, cells within a field of view often reside in slightly different focal planes (Fig. 4b; Supplementary Fig. 3).” Can the authors comment whether this could occur if the image requires a flat field correction, i.e., the microscope is not aligned correctly?

7) The lines 239-240 seem to undermine the main argument laid out in lines 78-81:

In lines 78-81, the authors explain that one motivation for MEDUSSA is that phase segmentation overestimates cell mask dimensions; yet in lines 239-240, they note that MEDUSSA’s FMSEg model also suffers from overestimation of mask sizes. The authors could reformulate these sentences to avoid dissonance in these arguments. Compare lines 78-81: “Despite its advantages, phase contrast images have limitations for cell size estimation. Firstly, they lack clear visual references to precisely define cell boundaries, which tends to result in an overestimation of cell size using currently available segmentation models (Hardo et al., 2024).” versus lines 239-240: “we observed that the FMSEg model consistently generated masks slightly wider than ground truth masks (Fig. 4d; Supplementary Fig. 4a).”

8) Lines 239-240:

Since MEDUSSA’s FMSEg model cell mask overestimation was fixed by a systematic correction based on Bayesian statistics, as described in lines 603-612 (material and methods “measurement transformation”), it is fair to ask whether the size overestimation obtained from the segmentation of phase images described in lines 78-81 could also be fixed by a systematic error correction? was this tested at some point?

9) Lines 313-315:

The authors explain the absence of filamentous cells in the stationary phase, stating that “These extreme elongation events were not observed in the stationary phase (Figure 5c), suggesting that it is a phenomenon associated with active growth and that these filaments septate into multiple cells or lyse upon entry into the stationary phase.” Since these observations were performed in agar pads containing FM4-64, could the absence of filaments be caused by the use of the FM4-64-containing agar pad itself, which likely applies some pressure on the cells? Could the filaments be found in stationary phase if the cells are taken directly from a liquid culture and imaged on a microscopy slide?

10) How sensitive is the method to slightly different concentrations of FM4-64 beyond the 0.5 µg/mL specified in the materials and methods “Microscopy” section?

11) Figures:

Figure 1: The intensity profiles of five cells are good representative data, but the authors should measure more cells for each microscopy method and provide a statistical test.

Figure 2: The bacteria and yeast images have very different cell sizes, yet they still segmented OK. Can the authors explain whether adjustments, such as changing kernel sizes or diameter model settings, were required to obtain cell segmentation with such size differences, or whether the images were resized before segmentation?

Figure 4d: The similarity of the distributions could quantitatively be compared using Kullback–Leibler Divergence, for instance.

Figure 6b and Sup Fig. 7b require statistical tests with p-values to quantitatively compare the distributions, and their differences or the lack thereof.

12) Computer code:

After creating the MEDUSSA environment and installing the requirements with pip, an error occurred during installation. The error description is attached as a .txt file.

13) Repository:

In the GitHub repository line “Then, on your terminal, run python, which will open the Python interpreter, there runt:” replace “runt” with “run”.

14) Relevant typos:

Lines 56 and 291: space missing between the references and the word “and”

Figure 6’s legend: has two “b’s” and no “c”.

Line 386: remove “or impossible”.

Figure 2. line 148: check for typo in the model’s name: “bact_fluor_omni”

Version 1:

Reviewer comments:

(Remarks to the Author)

Summary:

In their revised manuscript; Reyes-Matte et al. strongly improved the representation and discussion of their membrane segmentation and downstream analysis pipeline MEDUSSA. Additional experiments were performed to assess the effect of 2D projection on membrane segmentation accuracy and the fine-tuned Omnipose models were tested on additional bacterial species, showing robustness and broad applicability of the method. Further state-of-the-art DL networks were trained and tested for comparison, an improvement to the previous submission. The rationale behind selecting Omnipose for further downstream analysis is now represented in a comprehensible way, although foundation models such as Cellpose-SAM already provide high-quality segmentations on raw data (as stated in the manuscript). Limitations and contradictory results are now also honestly discussed in the manuscript. Advantages of the models trained for preprocessing are outlined comprehensively and an additional CARE model was trained to allow for robust segmentation of low signal-to-noise images. Models are provided via the Github repository, which provides sufficient information to run the segmentation and analysis pipelines. All data, code and models are shared via public repositories, which is a great contribution to open science. Overall, the authors did a good job in responding to the raised points and addressed the comments extensively. However, and despite the conducted control experiments, projection effects might still affect segmentation performance and cannot be ruled out. I would encourage the authors to perform simulations on realistic scales that reflect experimental data. Apart from this major point some minor questions remain that, in my opinion, can be addressed without any further experiments. At this point, I already want to thank the authors for the extensive revision and detailed answers to the reviewer's comments.

Major

1. Projection artifacts: The authors performed two experiments to rule out the projection artifacts that are described by Hardo et al., 2024: Firstly, they simulated membrane images of bacteria with varying size and analyzed the resulting stacks by FilamentJ. Secondly, PBP1 expression was tuned to adjust the diameter of bacteria experimentally. These cells express a cytosolic marker, whose segmentation is compared to the segmentation of the FM4-64 membrane stain. I appreciate the effort that the authors took but still have concerns regarding the simulations undertaken in this revision. The description in the methods indicates that cell outlines were blurred using theoretical PSF parameters plane-by-plane. However, it is not clear whether the analyzed plane contains 2D projection effects or whether a single plane was isolated for the segmentation. In a high magnification widefield system, typical projection volumes are in the range of 400 – 500 nm, representing 5-6 planes of the simulated stacks. If individual planes were analyzed, the effect of 2D projections would not be fully accounted for. A complete inclusion of projection effects would most likely lead to a broadening of the membrane signal towards the cell center in smaller cells as shown by Hardo et al., 2024 (Figure 2a). To shed light on the different results of this manuscript and the work of the Bakshi lab, simulations could be performed with the SyMBac package and analyzed by the approach presented here. Another reason for the absence (or small contribution) of projection effects could be the simulation parameters for the cell shape. The minimal simulated cell width in Suppl. Fig. 1 is 19 px. Assuming a pixel size of 65 nm (100x magnification captured on a PCO Edge; actual simulated pixel size is not provided) would result in a minimal simulated cell width of 1.235 μm . This value is > 1.5 fold higher than the observed width of *B. subtilis* cells analyzed in this work. The authors should at least go down to the observed width range, and ideally beyond (e.g. down to 500 nm width). I expect that projection effects will become more evident when going to realistic scales, and this has to be reflected in the manuscript. The second approach (Suppl. Fig. 14, R2R, point 8) nicely demonstrates that cell widths obtained from membrane or cytosolic signals strongly correlate across a large width range. While the data looks convincing, projection effects could be present in both channels and might cancel each other out. I want to emphasize that comparative studies with species/strains in a similar width range is well possible with the current approach/pipeline. However, statements about absolute cell width are only possible if projection effects are considered and corrected reliably.

Minor

1. Line 86: The authors state that phase contrast lacks precisely defined cell boundaries. They could refer to the examples in Fig. 1 e/f, where this is nicely demonstrated by the line profiles of phase contrast /membrane fluorescence
2. Lines 117+: The authors tested different models fine-tuned to the bacterial training dataset that they generated. It is not stated whether all models were trained until convergence. Was the training performed on 2D images or stacks?
3. Line 126/127: It might be worth mentioning that it was confirmed on simulated data
4. Line 153: Can the authors hypothesize why the Cellpose-SAM model trained on deconvolved images performs worse than the model trained on raw images? This finding is very surprising and raises the question whether the model was trained until convergence. The authors might want to clarify this.
5. Line 189: The authors state that FM2FM can be applied when direct deconvolution is not feasible? Can they specify which cases they refer to?
6. Line 195: I appreciate that the authors tested FM2FM on data acquired on different microscopes. The results look convincing. Were the image settings selected to provide images with similar pixel sizes and was the intensity set to a similar dynamic range? Such details might be interesting for the reader, as differences in such parameters can affect model performance significantly.
7. Line 231+: This section seems out of place and might benefit from a better transition. I guess MEDUSSA acts after the FM2FM segmentation described in the sections before and is a pure instance analysis tool. Is this correct?
8. Lines 294 – 300: The added discussion is valuable. However, the conclusion that these results indicate realistic size estimates seems to be in contrast to the 14% difference. This section might be rephrased to highlight the suitability for comparable/semi-quantitative measurements instead of absolute measurements.
9. Lines 322+: Differentiation between biological and technical variability (Suppl. Fig. 11) is sound and convincing.
10. Lines 331 – 337: Typical CV values for *B. subtilis*, as mentioned in the rebuttal, should be added to the section for

comparison

11. Lines 461-471: The added discussion is very valuable and represents current challenges in the field. One of the challenges is the small size of a bacterial cell and the relatively large effect of over- or under-segmentation by a single pixel. However, there are solutions for segmentation with sub-pixel accuracy (see e.g. MicrobeTracker: <https://pmc.ncbi.nlm.nih.gov/articles/PMC3090749/>). This should at least be mentioned, if not put into perspective for future developments.

12. Lines 580 – 583: Pixel sizes should be added for the different microscopes, as it is important to assess model generalizability.

13. Lines 607-616: Provision of training parameters is valuable. However, it is not clear whether the models converged. One option to assess training performance is to provide the loss curves for each model in the Supplementary Information. This might be particularly important to ensure a valid comparison between the models, as it is performed in Figure 2.

14. Figure 2a: The F1 score distributions of some models seem to be truncated (e.g. microSAM or Morphometrics trained on raw data). Why is this the case?

15. Figure 2D: The segmentations for yeast or cocci seem to underestimate cell size by segmenting the cell cytosol instead of the membrane peaks. Can the authors determine and provide the IoU for these species? Assessment of model performance is required to state that segmentations are high quality and reliable.

16. Fig. 4d: The width of *B. subtilis* cells in Figure 3 is much closer to the EM-extracted values. Can the discrepancy that was mentioned in the previous review be due to biological/technical variation of the replicate shown in Figure 4? On the other hand, Suppl. Fig. 11 shows reproducible widths of ~ 0.77 μm for different samplings.

17. Figure 6: The width distribution of the WH320 wildtype strain (as shown in Figure 5) should be added to the graph for comparison. This helps to underscore the qualitative assessment provided in lines 399-400.

18. Suppl. Fig. 4: Can the authors explain why MTG performs worse than QoxB? As both approaches label the membrane, I would have expected a very similar performance.

19. Suppl. Fig. 6d: The low p-values shown for some metrics might be due to the high sample size (“p-hacking”; <https://doi.org/10.1038/s41598-021-00199-5>).

20. Suppl. Fig. 9: The overview is indeed helpful but seems to be outdated. The authors should include the version that is provided in the readme file on the Github repository (I assume this was planned anyway).

21. Suppl. Fig. 12: Thanks for performing the requested analysis. The explanation provided by the authors is convincing.

22. Suppl. Fig. 14: Are the values for FM4-64 and GFP switched or does the cytosolic marker indeed provide larger segmentation masks than the membrane signal? I would have expected a larger cell width for the membrane marker. In lines 481-495, the authors refer to over-/under-segmentation, as suggested by Hardo et. al. It might be valuable to directly connect this discussion to the data provided in Suppl. Fig. 14. Additionally, contradicting information is provided regarding the fluorescent protein (mNeonGreen, GFP and mGreenLatern are used in the figure, caption and text).

Reviewer #2

(Remarks to the Author)

I co-reviewed this manuscript with one of the reviewers who provided the listed reports. This is part of the Communications Biology initiative to facilitate training in peer review and to provide appropriate recognition for Early Career Researchers who co-review manuscripts.

Reviewer #3

(Remarks to the Author)

The authors have addressed my previous concerns.

Reviewer #4

(Remarks to the Author)

The authors have address the all points in full regarding the possibility that lack of clustered segmentations in the training datasets could affect image predictions, the implementation of SSIMs to better assess image similarity in figs 3 and 6; they also have provide a better explanation for the pipeline used in the algorithm, strengthen conclusions by increasing sample sizes when measuring single cell size in Fig1 and quantitatively comparing distributions in fig4, furthermore, the authors now directly state the limitation of their methods and direct the reader to sup. fig 6 for addressing biases in cell size measurements.

In addition, providing new comparisons to the newer segmentation models (CellposeSAM), has increased the relevance of this work.

The article could be published in its current form once the authors ensure their software can be installed properly:

1. wget not available on macOS / Windows

The README uses wget, which is not available by default on macOS or Windows.

Use curl instead:

```
curl -L https://github.com/OReyesMatte/MEDUSSA/archive/master.zip -o master.zip
```

2. unzip issues on Windows

The unzip command may not work by default on Windows. You can use PowerShell:

```
Expand-Archive master.zip
```

3. Incorrect repository URL in README

The repository URL in the README is incorrect. The correct archive URL is:

```
git clone https://github.com/OReyesMatte/MEDUSSA.git
```

Recommended Approach for all Platforms

```
conda create -n medussa_env -c conda-forge python=3.12 numpy -y && conda activate medussa_env
git clone https://github.com/OReyesMatte/MEDUSSA.git
cd MEDUSSA
pip install -r requirements.txt
```

Then verify installation:

```
python
```

Inside Python:

```
from MEDUSSA.utils import InstallCheck
InstallCheck()
```

We thank the reviewers for their comments and criticism, which have helped us to substantially improve the manuscript. Below we reply to all their comments. Reviewer's comments are in black font, and our responses in blue font.

Referee expertise:

Referee #1/2: Advanced Imaging

Referee #3: Biological imaging, machine learning

Referee #4: Machine learning and microbial cell biology

Reviewers' comments:

Reviewer #1 (Remarks to the Author):

In their manuscript, Reyes-Matte et al. present a framework for deep learning (DL) based cell segmentation and downstream analysis of bacterial morphology. The proposed method is based on the Omnipose network and, according to the authors, provides robust cell segmentation across various cell shapes using standard membrane staining protocols. One problem of phase contrast microscopy is the inability to segment filamentous bacteria that consist of several individual cells not discernible in phase contrast images. The authors tested various “out-of-the-box” DL models dedicated to segment bacterial bioimages. The poor performance motivated them to develop their image analysis pipeline MEDUSSA. This pipeline used DL-based segmentation of deconvolved membrane-stained images and was used to compare cellular dimensions of various *Bacillus* and *Priestria* strains across different growth stages. Morphological analysis revealed an unexpected phenotype that the authors traced back to a point mutation in the PBP1 protein. Complementation assays verified that PBP1 expression increases cell diameter, while inactivation or deletion leads to the observed narrow-cell phenotype. In addition to the segmentation model, the MEDUSSA framework also includes two image restoration models that can be used to predict deconvolved images and infer a membrane stain from cytosolic labeling. The README file in the Github repository provides information about installation of the required packages and some information about model functions and parameters. Actual guidelines on how to use MEDUSSA properly on custom data are not provided, though. What we appreciate a lot is that code and data are shared publicly, making it a valuable resource for other researchers.

While the manuscript is well-written and data analysis was carried out carefully, the impact and relevance of the presented work is not evident. The performance of the MEDUSSA pipeline is not compared to new DL networks, in particular foundation models (e.g. CellposeSAM). Furthermore, it is not clear whether the presented pipeline extracts correct information, or whether the results are systematically affected by projection artifacts. Quantification of model performance could be improved, in particular for the image restoration networks. Finally, the results are not discussed critically but seem to be presented overly optimistic.

In the following, we will elaborate on major/minor issues and hope that our comments are helpful to the authors.

Major points:

1) CellposeSAM (as of May 2025) represents a significant advance over previous cell segmentation methods by robustly generalizing across imaging modalities, data qualities, and cell sizes, now outperforming both inter-human agreement and previous benchmarks for accuracy in most common conditions. Unlike classical U-Net-based models, its transformer backbone—adapted from the Segment Anything Model (SAM)—enables improved feature extraction and compositional generalization, supporting high precision segmentation even in challenging scenarios. Currently, there is no evidence that CellposeSAM systematically overestimates cell area in the same way that classical models sometimes do, especially on lower-quality or out-of-focus images.

To address this comment, as well as the first comment from Reviewer 3, we expanded the set of segmentation algorithms evaluated. In addition to Omnipose and Cellpose3, we included Cellpose-SAM, microSAM, and Morphometrics. The deep-learning-based models, Omnipose, Cellpose-SAM, microSAM, and Cellpose3, were fine-tuned for fluorescent membrane segmentation using our ground-truth dataset. For each model, we evaluated performance separately on raw membrane images and on deconvolved membrane images, using models fine-tuned on the corresponding image type. Cellpose-SAM gave the best segmentation performance on raw membrane images, followed by Omnipose, Morphometrics, Cellpose3, and microSAM (Revised Figure 2a). However, Cellpose-SAM did not reliably segment highly elongated cells (Revised Supplementary Fig. 2 and Revised Supplementary Fig. 3). We assessed this using images of a *Lysinibacillus* strain that naturally forms very long cells. In contrast, Omnipose handled elongated cells well, but performed worse in dense clusters (Revised Supplementary Fig. 2).

We then tested whether deconvolved membrane images could improve Omnipose performance on clustered cells. The Omnipose model fine-tuned on deconvolved membrane images showed improved segmentation of cells in clusters relative to the corresponding raw-image Omnipose model, whereas Cellpose-SAM fine-tuned on deconvolved membrane images still did not reliably segment elongated cells (Revised Supplementary Fig. 3). It is interesting to note that the effect of deconvolution was model-dependent (Revised Fig. 2a). Omnipose, microSAM, and Morphometrics performed better on deconvolved membrane images, whereas Cellpose-SAM and Cellpose3 performed better on raw membrane images. The Omnipose model fine-tuned for deconvolved membranes reached performance comparable to that of Cellpose-SAM on raw membrane images. Because of its high performance and its ability to segment long cells, we moved forward with the Omnipose model fine-tuned for deconvolved-membrane segmentation.

We have revised the Results section to introduce this expanded benchmarking and motivate our choice of the final workflow (Lines 118-157). We have modified manuscript Fig. 2a to include benchmarking of all the models, and introduced two new Supplementary Figures (S2 and S3) to illustrate these results.

Revised Figure 2. Evaluation of models for fluorescent-membrane segmentation. **a**, Quantitative benchmarking of Omnipose (pink), CellposeSAM (blue), microSAM (yellow), Cellpose3 (green) and Morphometrics (gray) on fifty-six images of *B. subtilis*, *B. thuringiensis* and *P. megaterium*. Omnipose, CellposeSAM, microSAM, and Cellpose3 were fine-tuned separately on raw and deconvolved fluorescent membrane images; models fine-tuned on raw images were evaluated on raw images, whereas models fine-tuned on deconvolved images were evaluated on deconvolved images. Top, F1 score across Intersection over Union (IoU) thresholds. Lines are the mean F1 scores of 56 images. Dashed lines correspond to model performance on raw fluorescent membrane images, and solid lines on deconvolved images. Bottom, F1 at IoU of 0.8. The first violin plot of each pair (lighter color and delineated by a dashed line) corresponds to model performance on raw images, and the second violin plot (darker color) to model performance on deconvolved images. The horizontal dashed line in the violin plots indicate the median, and the dotted lines the first and third quartiles. We called Omnipose model fine-tuned on deconvolved membranes FMSeg_omni (FMSeg) **b**, Examples of images showing chains, clusters, and heavily elongated cells. Top, deconvolved micrographs of *B. subtilis*, *B. thuringiensis* and *P. megaterium* cells stained with FM 4-64; bottom, FMSeg masks. F1 scores at IoU 0.8 are indicated at the top-right corner of each image. Scale bar, 15 μ m. **c**, FMSeg segmentation of a *B. subtilis* microcolony. The image is split into three sectors. Segmentation results are shown in the middle sector, and FM 4-64 stained membranes in the flanking sectors. Scale bar, 30 μ m. **d**, Segmentation of cells of different shapes and sizes, with membranes labeled with FM 4-64 or other fluorophores. Top, deconvolved fluorescent images; bottom, FMSeg segmentation. *Saccharomyces cerevisiae*, *Micrococcus sp.*, *Lactococcus lactis* and *B. pumilus* cells were stained with FM 4-64 (red). *B. subtilis* was stained with MitoTracker Green (MTG). We also imaged a *B. subtilis* strain producing a GFP fused to the membrane protein QoxB (QoxB-sfGFP, green), which is homogeneously distributed in the membrane. See Supplementary Fig. 4 for benchmarking of MTG and QoxB-GFP segmentation. Scale bars, 10 μ m.

a Segmentation examples of raw test data

b Segmentation of elongated *Lysinibacillus* cells

Revised Supplementary Figure 2. Examples of raw membrane segmentation with different algorithms. Omnipose, Cellpose-SAM, microSAM and Cellpose3 were fine-tuned for the segmentation of non-deconvolved (raw) fluorescent membrane images. **a**, Examples of model performance on images from the benchmarking set used to calculate the F1 scores shown in Fig. 2a. First column, micrographs of *B. subtilis*, *B. thuringiensis* and *P. megaterium* cells stained with FM 4-64; second column, ground truth segmentation masks obtained with JFilament; third to seventh columns, segmentation masks obtained with the corresponding models. Masks of individual cells are in different colors. White arrowheads in the Omnipose segmentation point at representative aberrant masks in regions of clustered cells. **b**, Examples of model performance on elongated cells from a natural *Lysinibacillus* isolate. First column, micrographs of long *Lysinibacillus* cells stained with FM 4-64; second to seventh columns, segmentation results from the different models. For Cellpose-SAM, we also tried resizing the images to half the original size [Cellpose-SAM (resized)]. Scale bars, 15 μm .

a Segmentation examples of deconvolved test data

b Segmentation of elongated *Lysinibacillus* cells

Revised Supplementary Figure 3. Examples of deconvolved membrane segmentation with different segmentation algorithms. Omnipose, Cellpose-SAM, microSAM and Cellpose3 were fine-tuned for the segmentation of deconvolved fluorescent-membrane images. **a**, Examples of model performance on images from the benchmarking set used to calculate the F1 scores shown in Fig. 2a. First column, deconvolved micrographs of *B. subtilis*, *B. thuringiensis* and *P. megaterium* cells stained with FM 4-64; second column, ground truth segmentation masks obtained with JFilament; third to seventh columns, segmentation masks obtained with the corresponding models. Masks of individual cells are in different colors. **b**, Examples of model performance on elongated cells from a natural *Lysinibacillus* isolate. First column, deconvolved micrographs of long *Lysinibacillus* cells stained with FM 4-64; second to seventh columns, segmentation results from the different models. For Cellpose-SAM, we also tried resizing the images to half the original size [Cellpose-SAM (resized)]. Scale bars, 15 μ m.

2) The paper should highlight that CellposeSAM sets a new standard for microscopy segmentation, with accuracy surpassing inter-human agreement and notable robustness to image degradation and biological variability. It does not introduce an area overestimation bias as has been observed in previous models, provided training and data alignment are performed appropriately.

We now talk about Cellpose-SAM and other transformer-based methods in the introduction (Lines 57-62). Performance assessment is shown in our reply to the previous comment. Regarding the comment that Cellpose-SAM does not introduce an area overestimation bias, we did not see any analysis in the Cellpose-SAM preprint that addresses this point.

3) What does MEDUSSA stand for? It seems to be an acronym, but it is not specified in the manuscript.

MEDUSSA stands for “MEMbrane Deconvolution and Segmentation for Size Analysis”. This was specified at the end of the result section “MEDUSSA, an analytical pipeline for extracting cell size from fluorescent membrane images.” (Lines 265-266 in the original submission). Now we also state this in the introduction (Line 69).

4) Lines 99-100: The authors state that they “trained a custom deep learning model to facilitate the segmentation of fluorescent-membrane images”. It is, however, not mentioned that this model is just a custom-trained Omnipose model instead of an extension or modification of a NN architecture. This should be mentioned in the Abstract and also throughout the manuscript, not only in the methods section. Training a published DL architecture on custom data is not a big deal nowadays and raises the question of whether this work is a significant contribution to the field.

We have modified the sentence as follows: “we have fine-tuned and benchmarked some of these [previously explained] models for segmenting fluorescent-membranes.” We have also modified the results to explain the fine-tuning and benchmarking of different segmentation algorithms, and indicate that we fine-tuned models for the segmentation of fluorescent membrane images in the abstract.

While we fine-tune and benchmark several pre-existing models for membrane segmentation, the main contribution of this manuscript is not simply retraining the models, but elaborating and testing a series of steps for extracting bacterial cell size, and providing a proof of principle for its application in comparative studies.

5) Figure S1: The comparison of selected segmentation approaches on membrane images of different bacterial species is useful. However, it is not clear whether optimal parameters were chosen for these models. For example, Omnipose requires rescaling of the data so that the cell width is in a suitable range. Has this been taken into consideration? Also, there are other methods that were shown to work on diverse morphologies and imaging modalities (e.g. Ursell et al., Morphometrics). Is there a reason why the authors did not test these models on the test datasets? We are aware of the focus on Deep Learning based methods, but if methods based on conventional algorithms or classical machine learning perform well, we would prefer those over DL methods due to better explainability.

The Figure S1 the reviewers refer to is no longer part of the manuscript. As explained in our response to comment 1 from these reviewers, we have now assessed the performance of multiple segmentation algorithms, including CellposeSAM and Morphometrics. Among the methods tested, fine-tuned Cellpose-SAM showed the best performance on raw membranes (although it

oversegmented long cells) and Omnipose showed the best overall performance on deconvolved membranes, and in addition was able to properly segment long cells.

Both CellposeSAM and Omnipose require an approximate cell size as input. However, both frameworks also provide an option to estimate cell size automatically rather than specifying it manually, and we used this automatic size-estimation mode in all cases.

6) Lines 162 – 170: What is the benefit of the image restoration model? It is expected that they perform similar to the method used to generate the ground truth (here deconvolution). Is the CARE prediction faster? If not, why not use deconvolution instead of a DL model that requires rigorous training and testing? Regarding the FP2FM model, would it not be straightforward to perform segmentation on the cytosolic marker? The cells appear to be well discernible by the cytosolic signal, even when they grow in chains (Fig. 3a, bottom panel). Moreover, the performances of these models were not quantified. Can the authors validate via suitable metrics (e.g. SSIM or RSME) that the predictions are artifact-free and high-quality?

In principle, nothing prevents researchers from applying conventional (“real”) deconvolution to their images (we now state this in the manuscript, lines 188-190). However, the image restoration model remains useful for three reasons. First, not all deconvolution algorithms are equivalent. By providing a model trained to restore images toward a specific deconvolution target (DeltaVision-deconvolved fluorescent membranes), we increase the likelihood that the output will be compatible with FMSeq and will yield consistent segmentation results. Second, the restoration model is part of our strategy to correct for focal-plane differences between cells within the same field of view (manuscript Fig. 4b-c and Supplementary Fig. 7) and is therefore a key component of the MEDUSSA pipeline. Third, the model approximates the output to deconvolved images of a widely used system in bacterial cell biology (DeltaVision), that has been recently discontinued. Because the proprietary implementation of DeltaVision deconvolution is becoming less accessible, the restoration model may facilitate comparability with existing datasets and help future users reproduce similar image characteristics without requiring the original platform.

Regarding the question about performing segmentation directly on the cytosolic marker rather than using the FM2FM model, this is certainly a possibility. The only issue is that cytosolic markers do not provide unambiguous borders for segmentation as do fluorescent membranes. But it would be possible to train a model on cytosolic fluorescence using masks obtained from fluorescent membranes.

For validation of the restoration model with SSIM, please see the reply to comment 2 from reviewer 4.

7) CARE can also be used to predict super-resolution images from widefield images (see DeepBacs, prediction of *E. coli* / *S. aureus* SIM images). Would such a network be more informative than predicting deconvolved images? As shown in Hardo et al. 2024, membrane stains in widefield imaging misestimate the underlying membrane position due to projection effects. This effect is actually shown in Fig. 4e, where the width determined in the deconvolved images systematically differs from the width of CryoEM data. Especially when it comes to volume comparisons, an underestimation of 15% at low cell widths can have a large effect on the volume ratio. This difference is not discussed in the manuscript, but instead the FMSeq segmentation is considered to be a “realistic cell size estimation” (line 260). However, this deviation is significant and should be discussed in detail.

CARE models for prediction of super-resolution images, such as that implemented in DeepBacs, can also be used. We compared cell-size outputs obtained from (i) conventionally deconvolved images (“real” deconvolution), (ii) images restored with FM2FM, and (iii) images restored with DeepBacsSR (see Response Figure 1, not included the manuscript). The results were similar in the three conditions. Predicted superresolution membrane images using DeepBacsSR did not result in wider cells, as might be expected if projection artifacts were prominent in our images. Instead, width measurements from images restored with DeepBacsSR were slightly smaller than those from deconvolved images or images restored with FM2FM. We now explicitly indicate that alternative restoration models, including those provided in DeepBacs, can also generate images that are suitable for segmentation with FMSeg (lines 222-223):

“Previously developed CARE restoration models for the prediction of superresolution membranes can also be used (Spahn et al., 2022).”

Figure 3. Distributions of cell width, length surface area and volume obtained from identical raw fluorescent membrane images processed in different ways: “real” deconvolution (Deconvolved Fm 4-64), restoration with FM2FM, and restoration with DeepBacs SR.

The reviewers appear to assume that the ~15% lower widths obtained from fluorescence microscopy images, compared to cryo-EM measurements (manuscript Fig. 4e), are due to projection artifacts. While this is one possible explanation, there are several alternatives. First, sample preparation for cryo-EM and fluorescence microscopy is different, which could affect cell width measurements. For example, for fluorescence microscopy (as done in our manuscript), cells are first grown in liquid culture and then placed in an agarose pad with fluorescent dyes, where they typically sit for 15-30 min before imaging. For CryoEM, cells grown in a liquid culture are placed in a carbon grill and plunged frozen before imaging. These differences in sample preparation might impact cell size. Second, fluorescent microscopy and CryoEM measurements were obtained from cells grown in different cultures, in different laboratories and with different batches of media. Although we keep media composition and sporulation conditions as consistent as possible, factors beyond our control might also cause differences in size between cultures. We now discuss this in the manuscript (lines 294-300):

" Our pipeline reproduced the difference in width between wild-type and $\Delta\textit{ponA}$ cells observed by cryo-EM, although the absolute width estimates were on average ~14% lower than those obtained by cryo-EM (Fig. 4e). This discrepancy may reflect a systematic bias in our fluorescence-based measurements, but it could also arise from differences in sample preparation, imaging modality, or growth conditions between the two datasets. These results indicate that our pipeline provides realistic cell size estimates and is robust to biologically relevant differences in cell width."

Beyond these potential alternative explanations, our results are also inconsistent with projection artifacts as the main explanation. Projection artifacts, as described by Hardo et al. (Hardo et al., 2024), are width-dependent and are expected to be more pronounced in thinner cells than in thicker cells. However, the width difference between fluorescent-membrane and cryo-EM measurements is essentially the same for thicker WT cells and the thinner cells of the *ponA* mutant. If projection artifacts were driving the discrepancy, we would expect a larger effect in the thinner *ponA* cells.

Nevertheless, it is worth noting that the pixel size in our fluorescence images corresponds to approximately ~10% of the cell width, which sets a practical limit on absolute accuracy. We acknowledge this as a general limitation of fluorescence-based width measurements (and therefore of our method) and have added a paragraph to the Discussion addressing this (lines 461-471).

8) Lines 239 to 249: In addition to the previous comment: The authors correct for a segmentation bias introduced by the model. However, it is not clear whether the ground truth annotation is actually accurate. Can the authors rule out that cell masks are underestimated during manual annotation, as suggested by Hardo et al., 2024? A benchmark with super-resolution or simulated data would be essential to validate that the "ground-truth" generated with FilamentJ can be considered as ground truth. This part is highly important, as a reliable cell size assessment can only be guaranteed by such benchmarks.

We have addressed this point using two complementary approaches. First, we used simulated data to test whether JFilament masks reliably represent cell dimensions. JFilament generated masks accurately reproduced the size of simulated rods across a range of sizes. We have added this analysis to the manuscript (Revised Supplementary Fig. 1).

Revised Supplementary Figure 1. JFilament snakes accurately delineate membrane position. **a**, Example of JFilament segmentation of a synthetic contour. JFilament snakes automatically adjust to and track the pixels of maximal intensity along a contour, generating a mask corresponding to the pixels enclosed by the contour. The top panel shows a JFilament snake (red) tracking a contour. The bottom panel shows the original contour (magenta) and the mask generated by JFilament (green). The mask excludes the contour pixel itself. **b**, Simulated rods of known sizes were used to generate fluorescent membrane signals with BlurLab (left) (Ursell et al., 2017), and the images were segmented with JFilament (right, JFilament masks). **c**, Rod widths measured from JFilament masks (y axis) versus true rod width (x axis). Width was estimated both by direct measurement of the mask (blue crosses; measure bounding box) and by applying the mask-based size calculation pipeline based on skeletonization described later in the manuscript (orange dots, skeleton). In both cases, the estimated widths matched the true widths closely. The diagonal black line marks a perfect correspondence between measured and true widths.

Second, we used a genetic approach to test whether projection artifacts bias cell-width estimates across the range of widths represented in our datasets in vivo. Hardo et al. described projection artifacts that have opposite effects depending on the fluorescent marker: membrane fluorescence tends to underestimate true width, whereas cytoplasmic fluorescence tends to overestimate it (Hardo et al., 2024). The magnitude of both biases was reported to be width-dependent, with proportionally larger effects in thinner cells (becoming very apparent in cells below $\sim 1 \mu\text{m}$) than in thicker cells. To probe for this effect under our imaging and analysis conditions, we constructed a *B. subtilis* strain that constitutively expresses a cytoplasmic fluorescent protein (mNeonGreen) and in which the sole *ponA* copy is placed under an IPTG-inducible promoter. In the absence of IPTG, cells are thinner, similar to a *ponA* mutant ($\sim 0.6 \mu\text{m}$ wide). As IPTG concentration increases, average cell width increases, reaching $\sim 1.2 \mu\text{m}$ at the highest IPTG concentration tested. We imaged both fluorescent membranes and cytoplasmic fluorescence, segmented each signal independently, and used the resulting masks to estimate cell widths (Revised Supplementary Fig. 14). If projection artifacts were prominent in our conditions, we would expect width estimates based on cytoplasmic fluorescence to be relatively larger than membrane-based estimates for thin cells, with the discrepancy decreasing as cells become thicker. However, we observe that width scales almost identically when estimated from membrane fluorescence and from cytoplasmic fluorescence across the entire width range. These results indicate that projection artifacts, as described by Hardo et al., do not bias our width estimates, at least within the tested range of cell widths.

We have added this as a supplementary figure (Supplementary Fig. 14) and refer to it in the discussion of the manuscript (lines 481-495).

Revised Supplementary Fig. 14. Projection artifacts do not bias width estimates across a range of cell widths. **a**, Representative micrographs of *B. subtilis* cells constitutively expressing cytoplasmic mNeonGreen and carrying the sole *ponA* copy under the IPTG-inducible Pspank promoter, grown at different IPTG concentrations. In the absence of IPTG, cells are thin, similar to a *ponA* mutant, whereas increasing IPTG progressively increases average cell width. Deconvolved FM 4-64 membrane fluorescence and cytoplasmic mNeonGreen fluorescence are shown. Scale bars, 10 μm . **b**, Cell width distributions estimated independently from membrane fluorescence (magenta; segmented with FMSeg) and cytoplasmic fluorescence (green; segmented with Cellpose-SAM) across cultures grown at different IPTG concentrations. This experiment was designed to test whether projection artifacts were affecting our measurements. Hardo et al. (Hardo et al., 2024) described projection artifacts that have opposite effects depending on the fluorescent marker: membrane fluorescence tends to underestimate true width, whereas cytoplasmic fluorescence tends to overestimate it. The magnitude of both biases was reported to be width-dependent, with proportionally larger effects in thinner cells (becoming very apparent in cells below 1 μm) than in thicker cells. If such effects were prominent under our imaging and analysis conditions, width estimates from cytoplasmic fluorescence would be expected to exceed membrane-based estimates more strongly in thin cells than in thick cells. Instead, the two measurements scaled similarly across the full width range. The upper graph shows absolute width values, and the lower graph shows normalized values, with mean widths for each segmentation mode set to 1 in the absence of IPTG. Absolute median widths were $\sim 5\text{--}10\%$ larger for mNeonGreen-based segmentations at all IPTG concentrations. Because masks were generated using different models for the two fluorescence signals, absolute widths are not expected to coincide exactly. However, the relative offset between the two measurements remained approximately constant across the tested width range, indicating that projection artifacts do not bias our width estimates, at least within the tested range of cell widths. At least 130 cells were measured per condition.

9) Line 282-283: The authors state that the results obtained by MEDUSSA are highly reproducible between replicates. However, the data shown in Supplementary Figure 6 show deviations of up to 15%. Is this deviation considered to be normal? The authors should discuss these deviations and also elaborate on whether they are of biological or analytical nature. Along these lines: Did the authors test different staining efficiencies? We assume that brightness of the membrane stain can affect the result, but this has not been assessed throughout this manuscript. Simulated data could be used to quantify the influence of different parameters (noise, brightness, ...) on segmentation quality and would allow comparison to the ground truth (see Hardo et al. 2024).

Most of the variation in cell width and length between experiments performed on different days is most likely of biological, rather than of technical nature. When we image samples from the same culture, we observe minimal differences in width and length (2-3% at most) (Revised Supplementary Fig. 11b). However, we sometimes observe larger differences when we image cells from different cultures. As explained in the reply to question number 7 from these reviewers, bacterial cell size is sensitive to culture conditions, and differences beyond experimental control can impact cell size. We now indicate this in the manuscript (lines 321-326):

" Single-cell dimensions were generally consistent across experimental runs, although we observed variation in population averages of some strains (up to 27% for cell length and 12% for cell width; Supplementary Fig. 11a) likely due to day-to-day differences in conditions outside experimental control. Technical variation among samples from the same culture was low (2-3% at most for cell width and length) (Supplementary Fig. 11b). "

a Size comparison in different experimental runs

b Sampling size does not affect cell size measurements

Revised Supplementary Figure 11. Length-width comparisons across experimental runs. **a**, Scatter plots show cell width (x-axis) versus length (y-axis) for individual cells from the six *P. megaterium* strains in Figure 5 (WH320, DSM 32, KM, QM B1551, DSM 319, and 899) and a *B. subtilis* strain (PY79). Each dot represents a cell, colored by experimental run (blue, Day 1; orange, Day 2). Exponential- and stationary-phase distributions are shown in separate panels for each strain. The inset in each panel reports the number of cells analyzed (n), the median width (W) and length (L), and their coefficients of variation (CV). Prior to measurement, segmentation output was manually curated to remove aberrant masks and truncated masks at image edges. **b**, Width (upper graph) and length (lower graph) distributions of *B. subtilis* cells from technical replicates from the same culture. The dashed line in the violin plots indicate the median, and the dotted lines the first and third quartiles. The number of cells analyzed and the mean width or length \pm standard deviation (in μm), are indicated above each violin.

We refer the reviewers to our response to comment 2 from reviewer 3 regarding the effect of staining efficiency on segmentation quality.

10) Hardo et al. show that the wavelength has a strong influence on the segmentation quality. Particularly fluorophores with emission at longer wavelengths increasingly underestimate the cell area. The authors should compare the segmentation quality of different fluorophores and investigate this effect in their data. Here, the data from the inner membrane GFP fusion (QoxB-sfGFP) could be compared to the FM4-64 stain.

We have compared cell width estimated from FM 4-64 and QoxB-GFP images, as the authors suggested (Revised Supplementary Fig. 12). We have found that width estimates are slightly smaller for QoxB-GFP than for FM 4-64 images (medians 0.71 μm and 0.74 μm , respectively) which is the opposite of what would be expected according to the wavelength effects proposed by Hardo et al. A likely explanation for our results is related to the position of the two fluorophores in the cell: FM 4-64 binds to the outer leaflet of the membrane, while the GFP is fused to a cytoplasmic domain of QoxB. Width estimations are therefore expected to be slightly smaller for QoxB-GFP than for FM 4-64.

Supplementary Fig. 12. Cell width estimation from FM 4-64 and QoxB-GFP images. Estimates were obtained from the same field of views, in which cells carrying QoxB-GFP were stained with FM 4-64. Over 80 cells were analyzed. The dashed line in the violin plots indicate the median, and the dotted lines the first and third quartiles.

11) Quantification of the performance of FMSeg on the species shown in Figure 2d would be of interest. Does the model perform equally well on the different datasets? Such quantification should be simple and rapid with the available code. It would also give partial answers to our previous comment.

We have quantified the performance of FMSeg on the segmentation of membranes stained with Mitotracker Green, and labeled with GFP fused to the membrane protein QoxB. Please see our reply to comment 4 from reviewer 3 for details.

12) Given our experience with DL models and particular CARE, the result of a CARE prediction is strongly dependent on the input image quality. Using data obtained under different optical configurations or microscope settings will eventually already result in artifactual reconstructions. This is even worse when applying a CARE model to data recorded on a different microscope. The authors should discuss how well the provided CARE model can generalize and showcase its performance on publicly available datasets of membrane-stained bacteria. This way it can be assessed whether the provided model can be used reliably or whether retraining is required when switching strains/settings/microscopes (we would expect the latter).

We used our CARE model to restore raw images acquired with four different microscopes (Revised Supplementary Fig. 5): a DeltaVision, which was used to generate the training images, a Leica Thunder (images taken by us), a Zeiss Axio Imager Z2 (images taken by us) and a Nikon Ti-eclipse (downloaded from an online repository (McKenzie et al., 2022)). The model produced images suitable for segmentation from data acquired on the three microscopes. However, we do expect restoration performance to depend on image quality. Low signal-to-noise images may not be properly processed. To account for this, we have trained an additional model using input images spanning a range of signal-to-noise ratios to predict a high-quality target. We refer the reviewers to our response to comment 2 from reviewer 3 for additional details.

Revised Supplementary Fig. 5. Restoration and segmentation of images acquired with different microscopes. Representative images of bacterial cells stained with FM 4-64, acquired using a DeltaVision, Leica Thunder, Zeiss Axio Imager Z2, or Nikon Ti-Eclipse microscope, restored with FM2FM, and segmented with FMSeg. The DeltaVision, Leica Thunder, and Zeiss Axio Imager Z2 images are from *B. subtilis* and were acquired in this study. The Nikon Ti-Eclipse images are from *E. coli* and were acquired by McKenzie and colleagues (McKenzie et al., 2022). Scale bars, 15 μm .

13) Figure 6: The results obtained from the complementation assays do not result in values comparable to the wild type strains. While we acknowledge that the observed trend matches the differences in cell width between the PBP1 variants of *Priestia megaterium*, we wonder what the reason for the discrepancy is (e.g. 0.9 μm width for WH320 vs. 0.7 μm in the fully induced complementation mutant). The authors should at least describe these deviations and provide possible explanations.

Bacterial cell width is multifactorial and depends on the combined activity of several protein complexes involved in cell-wall synthesis and central cellular physiology. In addition to class A PBPs (PBP1), the cell elongation machinery organized around MreB (the elongasome) plays an important role in setting cell diameter, together with other enzymes and regulatory factors that differ between species. Thus, expressing *P. megaterium* PBP1 in *B. subtilis* replaces only one component of a broader, species-specific cell-width control network. In addition, PBP1 is likely fine-tuned to the cellular context of its native organisms and may therefore function less efficiently in a heterologous host. Together, this explains why cell widths shift but do not fully reach *P. megaterium* values.

14) In general, the described PonA phenotype is rather dramatic with a width decrease of 35%. This difference does not require robust quantification as it is evident from brightfield images. Can the authors provide scenarios where subtle differences in cell dimensions need to be determined?

One possible application is to quantify the distribution of cell widths within a single culture (for example, to calculate the CV). Another is to distinguish subtle width changes caused by graded induction of proteins such as PBP1. Another application is to compare cell width across growth stages, for example between exponential and stationary phase, or two explore variation in cell width between biological replicates, as explained above, or between closely related strains. In all these scenarios, width differences may be small.

15) An overview on the image analysis pipeline of MEDUSSA would be useful. This could include the individual steps and alternative routes (e.g. whether and when CARE models are used for inference). Such a figure could be based on the overview provided in the README file of the Github repository. While reading the manuscript, it was actually not clear whether both models (FM2FM, FP2FM) are supported in the pipeline and whether you can choose to segment cells on non-deconvolved or deconvolved images.

Thank you for the suggestion. We have created an overview figure and have included it in the supplementary information (Revised supplementary Fig. 9).

Supplementary Fig. 9. Overview of the MEDUSSA pipeline. MEDUSSA predicts deconvolved fluorescent membrane images that are amenable to high-quality segmentation using custom CARE models. Starting from raw fluorescent membrane images (FM2FM), cytoplasmic fluorescence (FP2FM), or low signal-to-noise raw membrane images (FM2FM-HiSNR), the pipeline restores membrane signals to a common deconvolution target. MEDUSSA also corrects focal-plane differences between cells within the same field of view by projecting z-stacks and processing the projection with FM2FM. The restored membrane images are then segmented with the fine-tuned FMseg_Omni model to obtain instance segmentations of individual cells. Finally, MEDUSSA quantifies cell-size features and supports direct morphological comparisons across strains, including error-propagation analysis that can be used to transform the resulting measurements.

16) The manuscript seems to be overly optimistic about the results obtained. Despite several deviations (15% difference between replicates; 15% difference in cell width between MEDUSSA and CryoEM results), there is no critical discussion included. The authors should provide an unbiased and honest view on the results and also discuss the limitations of their algorithm. Particularly, the statement that MEDUSSA is applicable to virtually any bacterial species (line 409) is not justified given the small number of species analyzed in this work.

We agree. The revised version includes a critical assessment of the method in the discussion (lines 453-495). In addition to the species shown in this manuscript, we have applied MEDUSSA to more than 150 strain from 16 different species so far.

17) In general, the provided framework represents a collection of scripts and a combination of already published networks. This would hardly justify publishing the workflow as a novel approach, but rather represents a showcase of how to use a custom-trained Omnipose model with some downstream analysis. If there would be a dedicated software or plugin (napari, Fiji) together with a data post-processing module and a detailed documentation about proper usage, the framework would likely be more appealing to biologists that want to use the approach on their data.

The aim of the manuscript is to provide a validated workflow for cell-size measurement from fluorescent membrane images, and to illustrate how it can be used in comparative studies, including the identification of mutations underpinning cell-size changes among strains. The original submission already offered important insight for cell-size measurements based on fluorescent membranes, for example, highlighting the importance of focal plane and providing strategies to mitigate cell-to-cell focal-plane differences within the same field of view. It also provided biological insight into cell-size variation among strains of the same species, differences in morphological variation across growth stages, and mutations underpinning cell-width changes. In the revised version, we have added a more thorough evaluation of alternative segmentation approaches for fluorescent membrane images and have explicitly assessed the potential impact of projection

artifacts that could bias width estimates. We therefore believe that the manuscript offers more than a simple combination of previously published networks.

We agree that a dedicated FIJI plugin would be useful. However, CARE and Omnipose are already accessible to non-expert users through established FIJI plugins, and users can run the custom-trained models we provide using these plugins. We will provide clearer documentation in the GitHub repository. Developing and maintaining a standalone napari/FIJI plugin would be ideal, but is beyond the scope of the current study and outside our current software-development capacity. We view this as a natural direction for future work.

Minor points:

1) Multi-label models were shown to provide more accurate segmentations than models that rely only on background and foreground (see work from David van Valen). Have the authors tested such models or tried to implement a multi-label approach into their model?

We have not explored multi-label models. This might be a subject for future work.

2) Introduction: The authors state that there are a limited number of studies regarding exact cell sizes and morphological heterogeneity of bacterial species. While this might be true for interspecies comparisons, there has been work on mutant libraries that also cover a wide range of morphologies. While such work is present as references in the manuscript, it is not mentioned in the introduction (e.g. Morphometrics <https://doi.org/10.1186/s12915-017-0348-8>) or BiofilmQ (<https://doi.org/10.1111/mmi.15064>). The authors should introduce methods that were already shown to work on a range of different phenotypes and samples.

We now refer to studies that have measure cell size in mutant libraries of *E. coli* and *B. subtilis* (Lines 39-40).

3) Line 91: “lipophilic”

Fixed.

4) Lines 91/92: Different membrane stains exist that show altered performance on various species. Example is Nile Red that works much better on *Staphylococcus aureus* compared to the FM dyes. Other membrane markers exist, e.g. Mitotracker or fluorescent D-amino acids as cell wall precursors. The authors might want to extend their list of membrane markers beyond FM4-64. Reading further, alternative membrane labels were introduced in Lines 143/144). The authors might want to generalize labeling strategies already at this part of the manuscript.

We now also mention Nile Red in that paragraph. Other dyes are introduced later.

5) Lines 129 and 132: Do the authors refer to Supplementary Figure 1 instead of 2?

That paragraph refers to the segmentation of raw membranes, and the data is shown in Supplementary Figure 2.

6) Figure 4C: The blue line should be added to the plot legends for clarity.

Done.

7) Fig. 5: Strain identifiers should be added to all panels for clarity.

We have added the identifiers to the left panels of each row (b and d). We did not add it to panel c because they would interfere with the inset.

8) Lines 310 – 315: The authors elaborate on rare events, such as extremely long cells. What is the frequency such events occur at? It is not clear whether such ‘outliers’ describe relevant biological phenotypes or just represent stressed cells that occur spontaneously.

Elongated cells in *P. megaterium* DSM 32 are rare, but not negligible (more than 10% of the cells are longer than 10 μm in exponential phase). We agree that the biological significance of these elongated cells is currently unclear. However, these events are strain-specific. If they merely reflected spontaneous “stressed cells”, their frequency should be more uniform across strains grown under the same conditions. The fact that some strains are more prone to producing elongated cells suggests underlying genetic or physiological differences, which is itself biologically interesting.

More broadly, rare morphological states can have disproportionate ecological and evolutionary consequences, and one advantage of our approach is precisely that it captures such low-frequency events. For example, elongation in a small fraction of the population could reflect a bet-hedging strategy, potentially increasing resistance to protozoan predation at the cost of increased vulnerability to certain phages. Well-characterized phase-variation systems sometimes generate very small subpopulations (sometimes 1 in 1000 or less) yet strongly impact population survival in fluctuating environments (Cota et al., 2012; Cota et al., 2015). Therefore, tools that quantify cell morphology across large numbers of individual cells are important not only for describing mean phenotypes, but also for detecting rare states that may reveal meaningful biology.

9) Lines 415 – 417: Why should filamentation only be visible with membrane stains? Strongly elongated cells in brightfield or phase contrast images are usually considered to be filamentous, so the reason for this phenotype not being described for *Priestia megaterium* might be a lack of interest in this strain.

Elongated objects can be visualized in brightfield or phase-contrast microscopy. But a limitation of these methods is that they do not reveal septa, so it is not possible to determine whether an elongated object represents a single cell or a chain of multiple cells, as we illustrate in the first figure of the manuscript. As a result, a long shape observed in phase contrast can be misinterpreted as a single elongated cell when it is in fact several cells connected in a chain (a common growth mode for *Bacillus* and related genera). This is precisely why membrane dyes or other septum markers are important for accurate measurements of cell length and for identifying real elongation events.

10) Line 425: The authors mention that a CV of 0.15 is untypically high. Can the authors include literature that supports this statement? What are typical CV values that are considered “normal” or represent a “tight regulation” of cell width?

References for this were at the beginning of the paragraph (lines 420-421 in the original submission; line 513 in the revised manuscript). We have added relevant references also after the

statement in line 425. Previously reported CV for *B. subtilis* range from 0.05 to 0.081, similar to our results.

11) Figure S4: Caption: 3000 thousand

Fixed.

Reviewer #2 (Remarks to the Author):

I co-reviewed this manuscript with one of the reviewers who provided the listed reports. This is part of the Communications Biology initiative to facilitate training in peer review and to provide appropriate recognition for Early Career Researchers who co-review manuscripts.

Reviewer #3 (Remarks to the Author):

The paper proposes a complete workflow to accurately segment bacteria from fluorescent membrane images, and to extract size parameters e.g. width, volume, area for comparative analysis across different bacterial strains. The core aim of the paper it appears is to set a standard workflow for morphometric analyses.

Generally, the paper presents the subject well, and motivates the need for fluorescent membrane imaging vs the current practice of using phase-contrast, in that the latter fails to delineate individual bacteria when they are compacted. Therefore, what appears to be one long bacteria is actually a collection of smaller bacteria. The authors also made efforts to benchmark each step of the pipeline and I applaud them for the detail in measuring the size discrepancy.

Couple of comments:

1. It is not so obvious to me the Omnipose architecture and more importantly the code implementation is the best in 2025. I would encourage authors to also retrain cellpose3, check and retrain cellpose-SAM, micro-SAM to see if these would not actually be better. In particular we see the out-of-box generalization performance of cyto3 in Fig.2a is miles better than omnipose.

We have revised the manuscript to assess the performance of additional segmentation algorithms, including fine-tuned versions of Cellpose3, Cellpose-SAM and microSAM. We refer the reviewer to the response to the first comment of reviewers 1/2 for details.

2. The segmentation model appears to be trained on higher signal-to-noise ratio deconvolved images. How does the performance drop off with signal-to-noise ratio? with less optimal or high-resolution fluorescence imaging? is there a minimum required resolution?

We have tested this by acquiring images with different illuminations, resulting in different signal-to-noise ratios, and then calculating the segmentation quality for each image. Segmentation tends to be better with higher signal to noise ratios. At ratios below ~5 (in deconvolved images), the segmentation quality drops for some images, yet many still have F1 scores above 0.8 at IoU of 0.8 (Revised Supplementary Fig. 13).

To improve the segmentation quality of low signal-to-noise images, we have trained another CARE restoration model (FM2FM-hiSNR) that predicts high signal-to-noise deconvolved membranes across a wide range of signal-to-noise ratios of the original raw image. Briefly, we

acquired multiple images from each field with different illumination conditions, generating a range of images from low to high signal-to-noise ratios. We used images with different signal-to-noise ratios as input and deconvolved versions of high signal-to-noise images as target. Restoration with this new model improves the segmentation quality of low signal-to-noise images (Revised Supplementary Fig. 13). We also include this new restoration model in the manuscript.

a Representative micrographs

b Effect of restoration on F1 score

Revised Supplementary Fig. 13. Impact of signal-to-noise ratio on segmentation accuracy. **a**, Example segmentation of a low-signal-to-noise image. Micrographs are shown at the top and the corresponding segmentation results at the bottom. First column, raw (non-deconvolved) low-signal-to-noise image (Noisy raw image) segmented with Omnipose fine-tuned for raw membrane segmentation; second column, deconvolved image segmented with Omnipose fine-tuned for deconvolved membrane segmentation (FMSeg); third column, FM2FM-predicted image segmented with FMSeg; fourth column, image predicted with the CARE model FM2FM-HiSNR, trained to predict high-signal-to-noise deconvolved images from input images spanning a range of signal-to-noise ratios, segmented with FMSeg. F1 scores at IoU 0.8 are indicated at the top-right corner of each image. The last column shows a high-signal-to-noise version of the same image, acquired at higher light exposure, which served as the target for FM2FM-HiSNR restoration and was segmented with JFilament (ground truth). **b**, F1 scores at IoU 0.8 as a function of image signal-to-noise ratio for deconvolved fluorescent membrane images (deconvolved FM4-64) and for images restored with FM2FM-HiSNR. Each blue dot represents one image (fourty images in total). FM2FM-HiSNR restoration improves segmentation quality of low signal-to-noise images. The same set of raw images was used to generate both plots.

We have also tested the effect of image resolution by artificially reducing image size. Segmentation quality decreases as resolution decreases, but we still obtain fairly high F1 scores for images downsized to half of the original size. In contrast, the drop in F1 score is substantially sharper when images are reduced to one quarter of the original size (Response Fig. 2, not included in the manuscript).

Response Figure 2. Effect of image resolution on segmentation quality. Images were downsampled to 0.75, 0.5, or 0.25 of their original size and segmented with FMSeg. The F1 score for each example is shown in the top-left corner. For segmentation of downsampled images, we adjusted the Omnipose diameter parameter to the value indicated at the top of each image.

3. The model performance appears to be evaluated with 16 images. Is this small number sufficient to get a stable evaluation? How does the F1 score curve change as the number of images increases? It should presumably plateau. Did authors check this?

We have increased the evaluation dataset to 56 images (Revised Fig. 2a). The results are similar to those obtained with the smaller dataset of 16 images.

4. Authors claim the model trained on FM 4-64 membranes generalizes to other markers. However, there is no quantitative benchmarking to support this statement. What is the F1 score decrease?

We have benchmarked segmentation of two other markers: Mitotracker Green and the membrane protein QoxB fused to GFP. For Mitotracker Green, the F1 score is above 0.8 at IoU of 0.8. For QoxB-GFP it is higher than 0.9. We have added this to the revised manuscript (Revised Supplementary Fig. 4).

a Model performance

b Test data example

Revised Supplementary Fig. 4. FMSeg benchmarking on the segmentation of Mitotracker Green and QoxB-GFP fluorescence images. **a**, Left, F1 score over IoU thresholds for the segmentation of membranes stained with Mitotracker green (MTG, orange) or labeled with a QoxB-GFP fusion protein (blue). The lines represent the mean F1 scored of fifteen images. Right, F1 at IoU of 0.8. The white dots and vertical lines within the violin plots represent the means and standard deviations. **B**, Examples of segmentations of QoxB-GFP and MTG images from benchmarking data. Deconvolved micrographs (membranes), ground truth masks generated with JFilament (ground truth), and FMSeg segmentation results (FMSeg) are shown. Scale bars, 15 μm .

5. Since authors explored training of models to predict deconvoluted images, couldn't you also train models to predict from phase-contrast images the FM 4-64 marker to bypass membrane marker choice?

We tried to do this (Response Figure 3, not included in the manuscript). It is possible to predict lateral membranes from phase contrast images, but the prediction fails to include septal

membranes. This is probably because phase contrast images do not contain any features that can allow the inference of septum position.

Response Figure 3. Prediction of membranes from phase contrast images.

6. model performance evaluation is confusing and somewhat unfair. e.g. in Fig.2a the retrained omnipose FMSeg is compared with off-the-shelf Cyto3 and bact_fluor_omni models. The fair comparison is to retrain all alternative models on the same dataset, to really show that omnipose architecture is indeed the best one. Also the pretrained models should be explicitly labeled as pretrained in the figure panel. The FM 4-64 is clearly an out-of-distribution dataset, and I don't expect foundation models to maximize their potential out-of-the-box.

We agree. We now evaluate fine-tuned versions of different segmentation algorithms. See our reply to the first comment of reviewers 1-2 for details.

7. How did authors generate the ground-truth cell widths? Is this from manual cell outlines? Would this not be subject to annotator bias, and quality of the fluorescence imaging? Presumably, the widths is sensitive to skeletonization, and ground-truth widths shouldn't use the skeletons, a derivative measurement, so did authors manually draw widths? It would be good to be specific and detailed about this which is the main motivation of the paper.

We used a semi-automated active-contour tool (JFilament) that traces the peak of membrane intensity around each cell. This reduces annotation bias and is robust across a wide range of signal-to-noise ratios. We now provide more details in the Results section about this (Lines 123–127). Importantly, our ground truth corresponds to cell outlines (segmentation masks), not to widths drawn manually. Cell width values were computed afterwards from these outlines using the same automated measurement procedure applied throughout the manuscript. In particular, we did not manually place width measurements, and skeletonization (when used) is treated strictly as a downstream, deterministic step applied uniformly to both the JFilament-derived masks and the masks predicted by each segmentation method.

Reviewer #4 (Remarks to the Author):

In this work, Reyes-Matte et al. offer an insightful study of bacterial cell segmentation and the calculation of cell size from fluorescent images depicting cytoplasmic and cell-membrane-associated structures. The study is insightful because it showcases the sequential use of fluorescence microscopy, deconvolution via image-restoration methods (CARE), deep learning-driven image segmentation (Omnipose), and systematic error correction to extract physiologically relevant data, such as individual cell size. MEDUSSA provides an option to obtain accurate size measurements despite experimental heterogeneity in acquired images, such as differences in focal planes or morphological features not readily available in phase contrast microscopy, the most common microscopy regimen for bacterial segmentation. Although the study follows a clear logic and is well written, the authors should address the following points before publication:

1) Lines 129-132:

The authors state that the model “struggled with clustered cells, often producing indented or truncated masks (Supplementary Fig. S2).” This could happen because there were not enough images with clustered cells in the training data set, preventing the algorithm from generalizing across crowded conditions. What was the proportion of images with clustered cells in the training data set? Could the clustered segmentations have been improved by adding more clustered images to the training data set?

A substantial fraction of our training data consisted of clustered cells. If lack of clustered cell training data was the main problem, we would expect that the model trained on deconvolved images, which used the same training images but deconvolved instead of raw, would also perform poorly in clustered cells. We suspect that the problem resides in the increased blur in crowded areas of raw images. It is also interesting to note that deconvolution did not increase the performance of all models. Cellpose-SAM and Cellpose3 performed better on raw than on deconvolved images (see Revised Fig. 2a).

2) Lines 166-167:

To prove the following statement: “The fluorescence profiles of the predicted images closely matched those of deconvolved images (Fig. 3a,b), demonstrating the accuracy of the predicted deconvolution,” the authors should provide a quantitative measurement of image similarity, for instance, the calculation of the structural similarity index SSIM for the area of the image with the membranes.

We have calculated the structural similarity index measure between deconvolved fluorescent-membrane images and images predicted with FM2FM or FP2FM (Revised Fig. 3c and Revised Supplementary Fig. 6c). Median SSIMs were 0.87 and 0.80 for FM2FM and FP2FM predicted images, respectively.

Revised Figure 3. Deconvolution prediction. **a**, Example of deconvolution prediction with FM2FM. Top, input raw fluorescent membrane images (Raw FM 4-64); middle row, true deconvolved fluorescent membrane images (Deconvolved FM 4-64); bottom, FM2FM-predicted image. White squares mark regions zoomed in at right (Block 1 and Block 2). Diagonal dotted lines indicate profile traces in **b**. Scale bars: full images, 10 μm ; zoomed-in blocks, 5 μm . **b**, Fluorescence profiles across the dotted lines in Block 1 (top) and Block 2 (bottom). Light blue, raw FM 4-64; orange, deconvolved FM 4-64; burgundy, FM2FM prediction. **c**, Structural similarity index measure (SSIM) between FM2FM predicted images and deconvolved images. The distribution of SSIM values across 74 image crops is shown. **d**, Violin plots of cell width, length, surface area, volume, cross-sectional area, convex hull area, eccentricity and solidity calculated from deconvolved FM 4-64 images (orange) and FM2FM-predicted images (burgundy). The white dots and vertical lines within the violin plots represent the medians and standard deviations. P values from statistical comparisons between distributions (ANOVA or Kruskal test; see methods for details) are indicated in each panel. Twenty images (over 1000 cells) were processed and segmented with FM2Seg. See the Methods for size calculation details.

Revised Supplementary Fig. 6. Prediction of deconvolved membranes from cytoplasmic fluorescence. We trained a CARE model (FP2FM) to predict deconvolved membranes from cytoplasmic fluorescence. **a**, Example of deconvolution prediction with FP2FM. Top, input raw cytoplasmic fluorescence images (Cytoplasmic mVenusQ69M); middle row, true deconvolved fluorescent membrane images (Deconvolved FM 4-64); bottom, FP2FM-predicted image. White squares mark regions zoomed in at right (Block 1 and Block 2). Diagonal dotted lines indicate profile traces in **b**. Scale bars: full images, 10 μm ; zoomed-in blocks, 5 μm . **b**, Fluorescence profiles across the dotted lines in Block 1 (top) and Block 2 (bottom). Light blue, raw FM 4-64; orange, deconvolved FM 4-64; dark blue, FP2FM prediction. **c**, Structural similarity index measure (SSIM) between FP2FM predicted images and deconvolved images. The distribution of SSIM values across 74 image crops is shown. **d**, Violin plots of cell width, length, surface area, volume, cross-sectional area, convex hull area, eccentricity and solidity calculated from deconvolved FM 4-64 images (orange) and FP2FM-predicted images (dark blue). The white dots and vertical lines within the violin plots represent the medians and standard deviations. P values from statistical comparisons between distributions (ANOVA or Kruskal test; see methods for details) are indicated in each panel. Although the distributions are visually similar, statistically significant differences ($p < 0.05$) were found for cell length, surface area, volume, cross-sectional area, convex hull area, and eccentricity, suggesting that deconvolved membrane prediction from cytoplasmic fluorescence could bias cell size estimations. Twenty images (over 1000 cells) were processed and segmented with FMSeg. See the Methods for size calculation details.

3) Lines 188-191:

The multiple size measurements are highly correlated parameters (length, area, volume, etc). A more comprehensive assessment could be performed using morphological descriptors that are not entirely associated with mask length, such as eccentricity, solidity, and convex hull.

We have calculated eccentricity, solidity and convex hull area for masks generated from deconvolved images, and from FM2FM and FP2FM predicted images. See Revised Fig. 3d and Revised Supplementary Fig. 6d above. We refer to this in the results section (lines 214-215) ("We also compared shape descriptors not directly related to cell size, such as convex hull area, eccentricity and solidity.")

4) Line 191: The statement "The size distributions were nearly identical (Fig. 3c)" requires p-values or statistical tests to clearly assess all comparisons.

We have now performed statistical tests (see Revised Fig. 3d and Revised Supplementary Fig. 6d above). Size distributions obtained from deconvolved images and from images restored with FM2FM are not statistically different. However, many size distributions obtained from FP2FM-restored images show statistically significant differences, although the differences in the absolute values are small. We have moved the FP2FM model to supplementary information and have added a statement in the result indicating that it may bias size calculations (lines 223-225): "Although these approaches expand the applicability of the method, they may introduce biases on cell size measurements (Supplementary Fig. 6)."

5) Line 201-202: Provide a brief explanation of the algorithm in the text.

We have added a brief description in the text (lines 234-241).

6) Lines 210 – 211:

"However, due to minor aberrations in microscopy slides, cells within a field of view often reside in slightly different focal planes (Fig. 4b; Supplementary Fig. 3)." Can the authors comment whether this could occur if the image requires a flat field correction, i.e., the microscope is not aligned correctly?

Flat-field correction corrects illumination uniformities, but the focal-plane differences we refer to reflect axial height differences of the cells within the field of view. In our experience, these small z-offsets most commonly arise due to slight tilt of the coverslip or agarose pad, or local unevenness in pad thickness, so different cells can lie at slightly different distances from the objective even within a single image. Incorrect alignment can probably exacerbate these differences, but this effect can occur even when the microscope is well aligned.

7) The lines 239-240 seem to undermine the main argument laid out in lines 78-81: In lines 78-81, the authors explain that one motivation for MEDUSSA is that phase segmentation overestimates cell mask dimensions; yet in lines 239-240, they note that MEDUSSA's FMSeg model also suffers from overestimation of mask sizes. The authors could reformulate these sentences to avoid dissonance in these arguments. Compare lines 78-81: "Despite its advantages, phase contrast images have limitations for cell size estimation. Firstly, they lack clear visual references to precisely define cell boundaries, which tends to result in an overestimation of cell size using currently available segmentation models (Hardo et al., 2024)." versus lines 239-240: "we

observed that the FMSeg model consistently generated masks slightly wider than ground truth masks (Fig. 4d; Supplementary Fig. 4a).”

We have removed the statement “...tends to result in an overestimation of cell size using currently available segmentation models”, as this may not hold true for all the segmentation models that we now benchmark in the manuscript (see response to the first comment of reviewers 1/2).

8) Lines 239-240:

Since MEDUSSA’s FMSeg model cell mask overestimation was fixed by a systematic correction based on Bayesian statistics, as described in lines 603-612 (material and methods “measurement transformation”), it is fair to ask whether the size overestimation obtained from the segmentation of phase images described in lines 78-81 could also be fixed by a systematic error correction? was this tested at some point?

It is possible that size overestimation in phase-contrast images can be systematically corrected. In the original manuscript, we referred to size overestimation in the context of Omnipose segmentation. In the revised manuscript, however, we include additional segmentation algorithms for which it is not clear whether size estimates from phase-contrast images are overestimated. We therefore no longer refer to phase-contrast size overestimation and have not explored this point further.

9) Lines 313-315:

The authors explain the absence of filamentous cells in the stationary phase, stating that “These extreme elongation events were not observed in the stationary phase (Figure 5c), suggesting that it is a phenomenon associated with active growth and that these filaments septate into multiple cells or lyse upon entry into the stationary phase.” Since these observations were performed in agar pads containing FM4-64, could the absence of filaments be caused by the use of the FM4-64-containing agar pad itself, which likely applies some pressure on the cells? Could the filaments be found in stationary phase if the cells are taken directly from a liquid culture and imaged on a microscopy slide?

Cells were imaged in agarose pads, but cells were grown in liquid culture to exponential or stationary phase and were only transferred to the pads before imaging (FM 4-64 was present in the pad). Since exponential and stationary phase samples were processed in the same manner, and elongated cells were observed in exponential phase, it is unlikely that transfer to pads impacts the detection of elongated cells in stationary phase. Instead, we think that elongated cells lyse or septate before stationary phase.

10) How sensitive is the method to slightly different concentrations of FM4-64 beyond the 0.5 µg/mL specified in the materials and methods “Microscopy” section?

We have not played with FM 4-64 concentration, but with light exposure to assess performance on low signal-to-noise images. We refer the reviewer to our response to the comment 2 from reviewer 3 for details.

11) Figures:

Figure 1: The intensity profiles of five cells are good representative data, but the authors should measure more cells for each microscopy method and provide a statistical test.

We have added the profiles of additional cells (39 total, updated in Revised Figure 1). However, we are not sure what would be an appropriate test to compare these profiles.

Figure 1. Phase-contrast versus fluorescent-membrane segmentation. **a, b**, Phase-contrast (**a**) and fluorescent membrane images (**b**) of *B. subtilis* growing vegetatively. Membranes were stained with FM 4-64. The insets show zoomed-in views of the boxed regions; the red lines illustrate the paths along which intensity profiles are calculated in panels **e** and **f**. **c, d**, Segmentation masks for the phase-contrast (**c**) and fluorescent membrane (**d**) images shown in **a** and **b**. The phase-contrast image was automatically segmented with Omnipose (Cutler et al., 2022) and the fluorescent-membrane image was manually segmented with JFilament (Smith et al., 2010) (see also panel **g**). Every mask has a different color. **e, f**, Phase-contrast (**e**) and fluorescent membrane (**f**) intensity profiles of 39 cells, measured along a 24-pixel line perpendicular to their long axis, centered at midcell (illustrated in insets in panels **a** and **b**). Pixel intensities were normalized to the maximum intensity observed across the lines for each imaging modality. **g**, Example of manual fluorescent-membrane segmentation using the active contour software JFilament (Smith et al., 2010).

Figure 2: The bacteria and yeast images have very different cell sizes, yet they still segmented OK. Can the authors explain whether adjustments, such as changing kernel sizes or diameter model settings, were required to obtain cell segmentation with such size differences, or whether the images were resized before segmentation?

For these images, we set the diameter flag to 0, so Omnipose estimates a diameter value for each image individually.

Figure 4d: The similarity of the distributions could quantitatively be compared using Kullback–Leibler Divergence, for instance.

We have done the comparison (Revised Fig. 4d).

a Measuring rod-shaped cells

b Solving focal plane differences

c Size comparison between projected cells and their optimal focal plane

d Cell size correction

e Fluorescence and Cryo-Electron microscopies

Figure 4. Cell size estimation from fluorescent-membrane segmentation masks. a, Cell size calculation workflow. b, Representative z-slices from an optical-sectioning stack of a 14-cell *B. subtilis* chain stained with FM 4-

64. Different cells are in focus in different slices. Circled cells are analyzed in **c**. The last image is a maximum-intensity projection of whole stack, processed FM2FM, where all cells appear in focus. Scale bar, 15 μm . Full stack in Supplementary Fig. 3. **c**, Width profiles (solid blue lines) of the cells circled in **b** on different z-slices across the optical sectioning stack: Cell 1 (right graph), Cell 7 (middle graph) and Cell 14 (right graph). Width is maximized on the slice in which the cell is in focus (vertical dotted lines). Cell width calculated from projected images is indicated by horizontal dashed lines. Results were equivalent regardless of the projection type (maximum intensity, red dashed line; average intensity, orange dashed line; sum, purple dashed line). Cells were segmented using JFilament in every z-slice and projections with FM2FM. **d**, FM2FM masks are slightly wider than manual JFilament masks. Density plots show the width distribution of 1100 bacilli cells. The top plot compares JFilament masks (blue) to FM2FM masks (green). After transforming the data with our sampling strategy (transformed masks, bottom plot), the distributions align better. The similarity of the distributions was quantitatively compared using Kullback–Leibler Divergence (KL div, values in insets within the graphs). See Supplementary Fig. 8 and the Methods section for details. **e**, Widths of wild-type (WT, blue) and ΔponA (orange) cells from sporulating *B. subtilis* cultures, measured from deconvolved fluorescent-membrane images segmented with FM2FM (FM 4–64 deconvolved; 326 and 617 cells for WT and ΔponA , respectively), or from CryoEM images (24 and 51 cells for WT and ΔponA , respectively). Dashed lines mark the median: WT 0.70 μm (FM) vs 0.81 μm (cryo-EM); ΔponA 0.54 μm (FM) vs 0.63 μm (cryo-EM). FM widths were $\sim 14\%$ lower than cryo-EM, but the WT/ ΔponA ratio was nearly identical (1.300 vs 1.304; $<1\%$ difference). Cryo-EM images courtesy of E. Tocheva (ΔponA) and E. Villa & K. Khanna (WT).

Figure 6b and Sup Fig. 7b require statistical tests with p-values to quantitatively compare the distributions, and their differences or the lack thereof.

We have now performed statistical tests for Fig. 6b and report the results in the figure legend in the manuscript. For Supplementary Fig. 7b (now revised Supplementary Fig. 15), statistical analysis was not possible in the original version because the comparison was based on a single cell. To address this, we have now compared size metrics from our *E. coli* dataset (Supplementary Fig. 8) obtained using the skeleton and medial-axis approaches. We detected significant differences in cell length and cell width distributions, likely arising from the higher branching propensity of the medial axis relative to the skeleton. We now mention this in the Methods and explain it in greater detail in the figure legend.

a Example cell and corresponding representations

c Dealing with branched skeletons

b Cell size across representations

d Cell size differences between representations

e Medial axis can overbranch cell masks

Supplementary Figure 15. Details of the imaging pipeline. **a**, Comparison of skeleton and medial-axis representations. Both were computed from a binary mask of an example FM 4-64-stained cell; pixel-wise differences between the skeleton and medial axis are shown in the bottom right panel. Scale bar, 15 μm . **b**, Comparison of the local widths and total cell length calculated using the skeleton (blue) or the medial axis (orange). Although similar, the two representations show slight differences in their local values. **c**, Graph strategy for branching skeletons. Four candidate skeletons were generated and size metrics computed for each; a single value per metric was then taken as the median across candidates (values shown inset in the “Skeleton inlay” panel). Scale bar, 2.5 μm . **d**, Cell size quantification of *E. coli* cells (ground truth masks used in Supplementary Fig. 8). Blue violins are the values obtained with the skeleton and orange violins are the values obtained with the medial axis. P values are indicated between violins (obtained from either an ANOVA or Kruskal test, see methods for details). **e**, Skeleton and medial axis representations of representative cells. Medial axis representation tended to over-branch more than the skeleton representation, which might explain the differences in some of the estimated cell size parameters. We proceeded with the skeleton representation.

12) Computer code:

After creating the MEDUSSA environment and installing the requirements with pip, an error occurred during installation. The error description is attached as a .txt file.

There was an error in the available installer files calling a non-existing python package, this has been fixed.

13) Repository:

In the GitHub repository line “Then, on your terminal, run python, which will open the Python interpreter, there runt:” replace “runt” with “run”.

Done.

14) Relevant typos:

Lines 56 and 291: space missing between the references and the word “and”
Figure 6’s legend: has two “b’s” and no “c”.

Line 386: remove “or impossible”.

Figure 2. line 148: check for typo in the model’s name: “bact_fluor_omni”

Thank you for catching the typos. We have fixed them.

Refernces

- Cota, I., Blanc-Potard, A. B., & Casadesus, J. (2012). STM2209-STM2208 (opvAB): a phase variation locus of *Salmonella enterica* involved in control of O-antigen chain length. *PLoS ONE*, 7(5), e36863. <https://doi.org/10.1371/journal.pone.0036863>
- Cota, I., Sanchez-Romero, M. A., Hernandez, S. B., Pucciarelli, M. G., Garcia-Del Portillo, F., & Casadesus, J. (2015). Epigenetic Control of *Salmonella enterica* O-Antigen Chain Length:

- A Tradeoff between Virulence and Bacteriophage Resistance. *PLoS Genet*, 11(11), e1005667. <https://doi.org/10.1371/journal.pgen.1005667>
- Cutler, K. J., Stringer, C., Lo, T. W., Rappetz, L., Stroustrup, N., Brook Peterson, S., Wiggins, P. A., & Mougous, J. D. (2022). Omnipose: a high-precision morphology-independent solution for bacterial cell segmentation. *Nature Methods*, 19(11), 1438-1448. <https://doi.org/10.1038/s41592-022-01639-4>
- Hardo, G., Li, R., & Bakshi, S. (2024). Quantitative microbiology with widefield microscopy: navigating optical artefacts for accurate interpretations. *Npj Imaging*, 2(1), 26. <https://doi.org/10.1038/s44303-024-00024-4>
- McKenzie, A. M., Henry, C., Myers, K. S., Place, M. M., & Keck, J. L. (2022). Identification of genetic interactions with priB links the PriA/PriB DNA replication restart pathway to double-strand DNA break repair in *Escherichia coli*. *G3 (Bethesda)*, 12(12). <https://doi.org/10.1093/g3journal/jkac295>
- Smith, M. B., Li, H., Shen, T., Huang, X., Yusuf, E., & Vavylonis, D. (2010). Segmentation and tracking of cytoskeletal filaments using open active contours. *Cytoskeleton*, 67(11), 693-705. <https://doi.org/10.1002/cm.20481>
- Spahn, C., Gomez-de-Mariscal, E., Laine, R. F., Pereira, P. M., von Chamier, L., Conduit, M., Pinho, M. G., Jacquemet, G., Holden, S., Heilemann, M., & Henriques, R. (2022). DeepBacs for multi-task bacterial image analysis using open-source deep learning approaches. *Communications Biology*, 5(1), 688. <https://doi.org/10.1038/s42003-022-03634-z>
- Ursell, T., Lee, T. K., Shiomi, D., Shi, H., Tropini, C., Monds, R. D., Colavin, A., Billings, G., Bhaya-Grossman, I., Broxton, M., Huang, B. E., Niki, H., & Huang, K. C. (2017). Rapid, precise quantification of bacterial cellular dimensions across a genomic-scale knockout library. *BMC Biology*, 15(1), 17. <https://doi.org/10.1186/s12915-017-0348-8>

Reviewers' comments:

Reviewer #1 (Remarks to the Author):

Summary:

In their revised manuscript; Reyes-Matte et al. strongly improved the representation and discussion of their membrane segmentation and downstream analysis pipeline MEDUSSA. Additional experiments were performed to assess the effect of 2D projection on membrane segmentation accuracy and the fine-tuned Omnipose models were tested on additional bacterial species, showing robustness and broad applicability of the method. Further state-of-the-art DL networks were trained and tested for comparison, an improvement to the previous submission. The rationale behind selecting Omnipose for further downstream analysis is now represented in a comprehensible way, although foundation models such as Cellpose-SAM already provide high-quality segmentations on raw data (as stated in the manuscript). Limitations and contradictory results are now also honestly discussed in the manuscript. Advantages of the models trained for preprocessing are outlined comprehensively and an additional CARE model was trained to allow for robust segmentation of low signal-to-noise images. Models are provided via the Github repository, which provides sufficient information to run the segmentation and analysis pipelines. All data, code and models are shared via public repositories, which is a great contribution to open science. Overall, the authors did a good job in responding to the raised points and addressed the comments extensively. However, and despite the conducted control experiments, projection effects might still affect segmentation performance and cannot be ruled out. I would encourage the authors to perform simulations on realistic scales that reflect experimental data. Apart from this major point some minor questions remain that, in my opinion, can be addressed without any further experiments. At this point, I already want to thank the authors for the extensive revision and detailed answers to the reviewer's comments.

Major

1. Projection artifacts: The authors performed two experiments to rule out the projection artifacts that are described by Hardo et al., 2024: Firstly, they simulated membrane images of bacteria with varying size and analyzed the resulting stacks by FilamentJ. Secondly, PBP1 expression was tuned to adjust the diameter of bacteria experimentally. These cells express a cytosolic marker, whose segmentation is compared to the segmentation of the FM4-64 membrane stain. I appreciate the effort that the authors took but still have concerns regarding the simulations undertaken in this revision. The description in the methods indicates that cell outlines were blurred using theoretical PSF parameters plane-by-plane. However, it is not clear whether the analyzed plane contains 2D projection effects or whether a single plane was isolated for the segmentation. In a high magnification widefield system, typical projection volumes are in the range of 400 – 500 nm, representing 5-6 planes of the simulated stacks. If individual planes were analyzed, the effect of 2D projections would not be fully accounted for. A complete inclusion of projection effects would most likely lead to a broadening of the membrane signal towards the cell center in smaller cells as shown by Hardo et al., 2024 (Figure 2a). To shed light on the different results of this manuscript and the work of the Bakshi lab, simulations could be performed with the SyMBac package and analyzed by the approach presented here. Another reason for the absence (or small contribution) of projection effects could be the simulation parameters for the cell shape. The minimal simulated cell width in Suppl. Fig. 1 is 19 px. Assuming a pixel size of 65 nm (100x magnification captured on a PCO Edge; actual simulated pixel size is not provided) would result in a minimal simulated cell width of 1.235 μm . This value is > 1.5 fold higher than the observed width of *B. subtilis* cells

analyzed in this work. The authors should at least go down to the observed width range, and ideally beyond (e.g. down to 500 nm width). I expect that projection effects will become more evident when going to realistic scales, and this has to be reflected in the manuscript. The second approach (Suppl. Fig. 14, R2R, point 8) nicely demonstrates that cell widths obtained from membrane or cytosolic signals strongly correlate across a large width range. While the data looks convincing, projection effects could be present in both channels and might cancel each other out. I want to emphasize that comparative studies with species/strains in a similar width range is well possible with the current approach/pipeline. However, statements about absolute cell width are only possible if projection effects are considered and corrected reliably.

We have performed the requested experiment: we simulated rods of different widths using SyMBac, applied a PSF to the 3D object, and then extracted the middle focal plane for analysis. Using this approach, JFilament-generated masks showed a close agreement with the true object width even for thin rods 400 nm wide (Response Figure 1). No obvious bias was observed. We have replaced Supplementary Figure 1 with this new figure.

Supplementary Figure 1. JFilament snakes accurately delineate membrane position. **a**, Example of JFilament segmentation of a synthetic contour. JFilament snakes automatically adjust to and track the pixels of maximal intensity along a contour, generating a mask corresponding to the pixels enclosed by the contour. The top panel shows a JFilament snake (red) tracking a contour. The bottom panel shows the original contour (magenta) and the mask generated by JFilament (green). **b**, Simulated rods of known sizes were used to generate fluorescent membrane signals with SyMBac (left) (Hardo et al., 2022), and the images were segmented with JFilament (right, JFilament masks). **c**, Rod widths measured from JFilament masks generated from mid-focal planes of SyMBac-simulated fluorescent membrane images, rescaled to a pixel size of 65 nm (measured width, y-axis), are plotted against the true widths of the simulated rods (true width, x-axis). Width was estimated by direct measurement of the JFilament-generated masks (black dots). Measured and true widths showed close agreement. The blue diagonal indicates perfect correspondence between measured and true values. R, Pearson correlation coefficient.

As the reviewers point out, the approach we used differs from that used by Hardo et al., in that we extract only the middle focal plane, whereas Hardo et al. project fluorescence from the full rod volume into a single plane. This difference could account for the different results obtained. Using Hardo's approach, the lateral membranes of cells $\sim 0.6 \mu\text{m}$ wide or thinner cannot be resolved (see Figure 2i in (Hardo, Noka et al. 2022)). While this is useful for illustrating that projection artifacts can affect measurements, it is important to note that lateral membranes of cells in this size range can in practice be readily resolved by standard epifluorescence microscopy. For example, in fluorescent-membrane images of sporulating *B. subtilis ponA* cells, which have an average width of $0.63 \mu\text{m}$ according to cryo-EM measurements (Figure 4e from our manuscript), the lateral

membranes are clearly resolved (Response Figure 2). Discussing and dissecting the reasons for the discrepancy between simulated and real microscopy falls beyond the scope of the present paper.

Response Figure 2. Raw (non-deconvolved) fluorescent-membrane images of a sporulating *B. subtilis ponA* mutant. Intensity profiles across cell width are shown in the right.

We disagree with the reviewers' suggestion that, in our *in vivo* experiments tuning cell width through inducible *ponA* expression, projection effects might be present in both channels and therefore cancel each other out. If projection artifacts were present at relevant magnitudes, those affecting membrane fluorescence and cytosolic fluorescence would be expected to reinforce rather than cancel one another. This is consistent with the analysis of (Hardo, Li et al. 2024) (again, see Figure 2i). The fact that width estimates obtained from membrane and cytosolic signals are strongly correlated across a broad width range strongly suggests that projection artifacts do not bias our *in vivo* results, at least within the range of widths examined here.

That said, we do not deny the potential importance of projection artifacts, as we already state in the Discussion (Lines 492-495): “These results suggest that projection artifacts, as described by Hardo et al., do not measurably bias our width estimates under these conditions, although they may become more relevant in cells thinner than those examined here or under different imaging settings.” We therefore think that conclusions about absolute width differences are justified within the width range examined in this manuscript.

Finally, we would like to point out that deconvolution may also reduce the impact of projection artifacts (see Figure 2f in (Hardo, Li et al. 2024)). This may contribute to the absence of appreciable projection artifacts in our *in vivo* experiments and provides an additional justification for the use of deconvolved membrane images in bacterial cell size studies.

Minor

1. Line 86: The authors state that phase contrast lacks precisely defined cell boundaries. They could refer to the examples in Fig. 1 e/f, where this is nicely demonstrated by the line profiles of phase contrast / membrane fluorescence

Done

2. Lines 117+: The authors tested different models fine-tuned to the bacterial training dataset that

they generated. It is not stated whether all models were trained until convergence. Was the training performed on 2D images or stacks?

Training was performed on 2D images. Models were trained following the recommendations in their respective documentations, as described in the Methods section of our manuscript. With the exception of microSAM, models were trained using the command-line interface (CLI), and convergence was assessed qualitatively by monitoring changes in the loss values throughout training. In all cases, loss variation was negligible during the final epochs, which we interpreted as evidence of convergence. microSAM models were trained using Jupyter notebooks, and the selected parameters corresponded to the epoch with the best performance.

3. Line 126/127: It might be worth mentioning that it was confirmed on simulated data

We refer readers to Supplementary Figure 1, where we show the simulated data.

4. Line 153: Can the authors hypothesize why the Cellpose-SAM model trained on deconvolved images performs worse than the model trained on raw images? This finding is very surprising and raises the question whether the model was trained until convergence. The authors might want to clarify this.

Please see our response to comment 2 for the convergence argument. As for why CellposeSAM performs worse on deconvolve images than in raw images, even after fine-tuning on each image type, this may reflect a reduced compatibility between the image features emphasized by deconvolution and the cues that CellposeSAM uses most effectively for segmentation. In particular, deconvolution replaces the smooth, continuous object-level appearance of raw membrane images with thinner, sharper, and more locally heterogeneous boundary signals that may be intrinsically harder for CellposeSAM to segment robustly.

5. Line 189: The authors state that FM2FM can be applied when direct deconvolution is not feasible? Can they specify which cases they refer to?

When we wrote this, we were thinking of images acquired with microscopes that do not have deconvolution capabilities, or images acquired from repositories. In these cases, system-optimized deconvolution cannot be applied, and therefore CARE models can be more useful.

6. Line 195: I appreciate that the authors tested FM2FM on data acquired on different microscopes. The results look convincing. Were the image settings selected to provide images with similar pixel sizes and was the intensity set to a similar dynamic range? Such details might be interesting for the reader, as differences in such parameters can affect model performance significantly.

Intensities were set to the same dynamic range (16bit). Excitation and emission wavelengths were kept as similar as possible while accounting for hardware differences between the three systems, as they all have their specific fluorescence cubes. Pixel size is now included in Supplementary Figure 5.

7. Line 231+: This section seems out of place and might benefit from a better transition. I guess MEDUSSA acts after the FM2FM segmentation described in the sections before and is a pure instance analysis tool. Is this correct?

MEDUSSA refers to the full pipeline, from segmentation to analysis. This is illustrated in Supplementary Figure 9. We have clarified this in lines 304-306: " We refer to the full pipeline of

deconvolution prediction, segmentation and cell size measurement as MEDUSSA (MEmbrane DeconvolUtion and Segmentation for Size Analyses) (Supplementary Fig. 9)."

8. Lines 294 – 300: The added discussion is valuable. However, the conclusion that these results indicate realistic size estimates seems to be in contrast to the 14% difference. This section might be rephrased to highlight the suitability for comparable/semi-quantitative measurements instead of absolute measurements.

We have modified the last sentence of the paragraph as follows:

" These results indicate that our pipeline captures biologically relevant differences in cell width and yields absolute size estimates that are broadly consistent with cryo-EM."

9. Lines 322+: Differentiation between biological and technical variability (Suppl. Fig. 11) is sound and convincing.

Thank you.

10. Lines 331 – 337: Typical CV values for *B. subtilis*, as mentioned in the rebuttal, should be added to the section for comparison

Added.

11. Lines 461-471: The added discussion is very valuable and represents current challenges in the field. One of the challenges is the small size of a bacterial cell and the relatively large effect of over- or under-segmentation by a single pixel. However, there are solutions for segmentation with sub-pixel accuracy (see e.g. MicrobeTracker: <https://pmc.ncbi.nlm.nih.gov/articles/PMC3090749/>). This should at least be mentioned, if not put into perspective for future developments.

This is an interesting point for future development. However, at this point we do not see a clear strategy for producing subpixel masks using DL methods.

12. Lines 580 – 583: Pixel sizes should be added for the different microscopes, as it is important to assess model generalizability.

The pixel sizes are now included in Supplementary Figure 5.

13. Lines 607-616: Provision of training parameters is valuable. However, it is not clear whether the models converged. One option to assess training performance is to provide the loss curves for each model in the Supplementary Information. This might be particularly important to ensure a valid comparison between the models, as it is performed in Figure 2.

Please see our response to comment 2 for the convergence argument. The loss curves were monitored during training, but way training was set up for these models does not provide an easy access them.

14. Figure 2a: The F1 score distributions of some models seem to be truncated (e.g. microSAM or Morphometrics trained on raw data). Why is this the case?

Violin plots were truncated to represent only the range where there's actual data points, instead of smoothed distributions.

15. Figure 2D: The segmentations for yeast or cocci seem to underestimate cell size by segmenting the cell cytosol instead of the membrane peaks. Can the authors determine and provide the IoU for these species? Assessment of model performance is required to state that segmentations are high quality and reliable.

We have only a limited number of images of cocci, and therefore cannot address this point in a meaningful manner without additional experimentation. In the manuscript, we present extensive benchmarking for rod-shaped cells, which are the primary focus of the study.

16. Fig. 4d: The width of *B. subtilis* cells in Figure 3 is much closer to the EM-extracted values. Can the discrepancy that was mentioned in the previous review be due to biological/technical variation of the replicate shown in Figure 4? On the other hand, Suppl. Fig. 11 shows reproducible widths of $\sim 0.77 \mu\text{m}$ for different samplings.

These differences likely reflect biological variation between samples.

17. Figure 6: The width distribution of the WH320 wildtype strain (as shown in Figure 5) should be added to the graph for comparison. This helps to underscore the qualitative assessment provided in lines 399-400.

That assessment refers to the ΔponA mutant that is already in the figure. The objective of this figure is not to replicate the cell size of WH320, but to show that there is an effect of the amino acid substitution on PBP1A function.

18. Suppl. Fig. 4: Can the authors explain why MTG performs worse than QoxB? As both approaches label the membrane, I would have expected a very similar performance.

Bacterial cells stained with MTG display a membrane-like signal. However, MTG is also known to bind proteins and may therefore produce more diffuse labeling than QoxB-GFP. This difference in labeling pattern may help explain the difference in performance.

19. Suppl. Fig. 6d: The low p-values shown for some metrics might be due to the high sample size ("p-hacking"; <https://doi.org/10.1038/s41598-021-00199-5>).

Thank you for the reference. This may explain the low p-values for some of our comparisons. We will take this into account in our future work.

20. Suppl. Fig. 9: The overview is indeed helpful but seems to be outdated. The authors should include the version that is provided in the readme file on the Github repository (I assume this was planned anyway).

We wanted to provide a simplified schematic of the general workflow.

21. Suppl. Fig. 12: Thanks for performing the requested analysis. The explanation provided by the authors is convincing.

Thank you.

22. Suppl. Fig. 14: Are the values for FM4-64 and GFP switched or does the cytosolic marker indeed provide larger segmentation masks than the membrane signal? I would have expected a larger cell width for the membrane marker. In lines 481-495, the authors refer to over-/under-segmentation, as suggested by Hardo et. al. It might be valuable to directly connect this discussion to the data provided in Suppl. Fig. 14. Additionally, contradicting information is provided regarding the fluorescent protein (mNeonGreen, GFP and mGreenLatern are used in the figure, caption and text).

The cytosolic marker produced larger masks than the membrane signal. However, it is important to note that the two signals were segmented using different models: CellposeSAM for the cytosolic marker and FmSeg_omni for the membrane signal. FmSeg_omni was trained on JFilament-annotated masks, which follow the membrane intensity peak around the cell, whereas the model used for the cytosolic signal may have been trained on manually annotated masks. In addition, cytosolic fluorescence does not provide unambiguous cues for defining cell boundaries in the same way as membrane fluorescence. This difference in training data and segmentation approach could explain the differences we observe.

As for the fluorescent proteins, we used ,GreenLatern. We have modified the figure and the legend to be more consistent. We thank the reviewers for catching this.

Reviewer #2 (Remarks to the Author):

I co-reviewed this manuscript with one of the reviewers who provided the listed reports. This is part of the Communications Biology initiative to facilitate training in peer review and to provide appropriate recognition for Early Career Researchers who co-review manuscripts.

Reviewer #3 (Remarks to the Author):

The authors have addressed my previous concerns.

Reviewer #4 (Remarks to the Author):

The authors have address the all points in full regarding the possibility that lack of clustered segmentations in the training datasets could affect image predictions, the implementation of SSIMs to better assess image similarity in figs 3 and 6; they also have provide a better explanation for the pipeline used in the algorithm, strengthen conclusions by increasing sample sizes when measuring single cell size in Fig1 and quantitatively comparing distributions in fig4, furthermore, the authors now directly state the limitation of their methods and direct the reader to sup. fig 6 for addressing biases in cell size measurements.

In addition, providing new comparisons to the newer segmentation models (CellposeSAM), has increased the relevance of this work.

The article could be published in its current form once the authors ensure their software can be installed properly:

1. wget not available on macOS / Windows
The README uses wget, which is not available by default on macOS or Windows.
Use curl instead:
curl -L <https://github.com/OREyesMatte/MEDUSSA/archive/master.zip> -o master.zip

2. unzip issues on Windows

The unzip command may not work by default on Windows. You can use PowerShell:
Expand-Archive master.zip

3. Incorrect repository URL in README

The repository URL in the README is incorrect. The correct archive URL is:
git clone <https://github.com/OREyesMatte/MEDUSSA.git>

Recommended Approach for all Platforms

```
conda create -n medussa_env -c conda-forge python=3.12 numpy -y && conda activate medussa_env  
git clone https://github.com/OREyesMatte/MEDUSSA.git
```

```
cd MEDUSSA  
pip install -r requirements.txt
```

Then verify installation:

```
python
```

```
Inside Python:  
from MEDUSSA.utils import InstallCheck  
InstallCheck()
```

Thank you for the suggestions. We have updated the instructions in the repository and tested them.

References

Hardo, G., R. Li and S. Bakshi (2024). "Quantitative microbiology with widefield microscopy: navigating optical artefacts for accurate interpretations." Npj Imaging 2(1): 26.

Hardo, G., M. Noka and S. Bakshi (2022). "Synthetic Micrographs of Bacteria (SyMBac) allows accurate segmentation of bacterial cells using deep neural networks." BMC Biol **20**(1): 263.